# Physically-Guided Data-Space Rectified Flow for Precipitation Nowcasting

Wenjie Luo [1 2]   Chaorong Li [1]   Chuanhu Deng [1 2]   Zhuo Wang [1 2]

## Abstract

Reliable long-horizon precipitation nowcasting requires preserving fine-scale echo structures while maintaining coherent transport. Although Rectified Flow (RF) can generate detail-preserving future sequences, numerical ODE integration compounds velocity estimation errors and induces progressive off-manifold drift, causing morphological distortions at extended lead times. We propose Physically-guided Data-space Rectified Flow (PDRF), which re-parameterizes the generative ODE in data space: the network predicts the clean future sequence, analytically inducing a coupled vector field with an implicit restoring effect that suppresses drift. To further enforce kinematic coherence, we introduce a soft Semi-Lagrangian teacher based on an advection prior to regularize large-scale transport, while allowing local growth/decay/deformation to be learned from data. Experiments on four public benchmarks demonstrate consistent improvements in event-based skill and better preservation of intense-echo morphology over long horizons.

## 1. Introduction

Accurate precipitation nowcasting—predicting high-resolution rainfall fields minutes to hours ahead—is pivotal for flood mitigation, emergency response, and infrastructure management (Wilson et al., 1998; Chen et al., 2025a; Li et al., 2025b). From a machine learning perspective, this task represents a challenging spatiotemporal generation problem characterized by high dimensionality and complex dynamics (An et al., 2025; Agrawal et al., 2025). Precipitation evolution involves two distinct physical regimes:

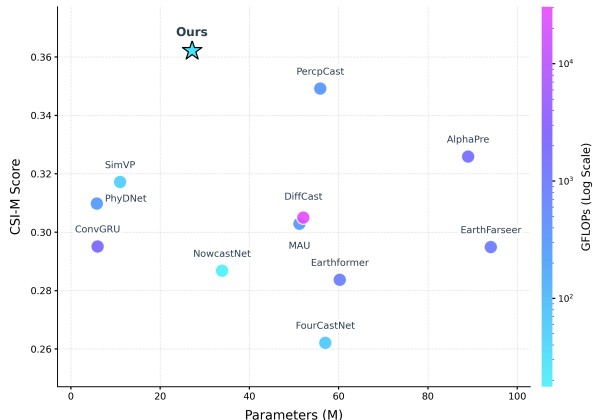

*Figure 1.* **Comparisons of model complexity and performance on the SEVIR dataset.** The scatter plot shows the CSI-M score (y-axis) versus the number of parameters in millions (x-axis). The marker color represents the computational cost (GFLOPs) on a logarithmic scale, ranging from blue (lower cost) to red (higher cost).

globally coherent advection (transport) and locally nonlinear intensity evolution (growth, decay, and deformation) (Seed, 2003; Bowler et al., 2006; Ha & Lee, 2024). As the forecast horizon extends, predictability degrades rapidly; small initial uncertainties can amplify into large-scale divergent futures (Germann & Zawadzki, 2002; Meo et al., 2024). Consequently, a robust nowcasting model must not only generate physically plausible snapshots but also maintain trajectory consistency over long horizons, ensuring that the generated sequence adheres to valid physical transport laws rather than succumbing to accumulation errors (Gong et al., 2025; Cao et al., 2025).

Existing approaches broadly fall into three categories. Classical extrapolation-based methods (e.g., optical flow) possess strong physical inductive biases for advection but struggle to model nonlinear intensity changes, leading to performance degradation when storm cells evolve rapidly (Bowler et al., 2006; Imhoff et al., 2020). Deterministic deep learning predictors trained with reconstruction objectives can capture nonlinear dynamics but inherently regress to the mean of possible futures. This results in blurred predictions that underestimate extreme events—precisely those most critical for disaster warning. To resolve this over-smoothing, generative forecasters have emerged as a powerful paradigm

[1]School of Computer Science and Technology (School of Artificial Intelligence), Yibin University, Yibin 644000, China [2]College of Computer Science and Engineering, Chongqing University of Technology, Chongqing 404100, China. Correspondence to: Chaorong Li <lichaorong88@163.com>.

to model uncertainty and sample detailed textures (Ho et al., 2020; Song et al., 2020; Albergo & Vanden-Eijnden, 2022). However, maintaining long-term coherence remains elusive: iterative generation often decouples temporal consistency, where small step-wise inconsistencies accumulate into trajectory-level drift, causing echo fragmentation and morphological distortion over extended lead times.

Rectified Flow (RF) (Liu et al., 2022) offers a promising avenue for efficient generation via deterministic ODE integration. Yet, applying standard RF to physical forecasting reveals a fundamental limitation. Standard formulations parameterize the network to regress a flow-space velocity field. In high-dimensional physical spaces, this velocity represents a latent derivative that is only indirectly tied to pixel-space validity constraints. During recursive ODE integration, inevitable regression errors in the velocity field compound, gradually steering the sampling trajectory away from the valid radar data manifold (Li & He, 2025). This geometric instability, termed off-manifold drift, manifests as the loss of structural integrity in long-horizon forecasts.

To address this failure mode, we propose a novel **P**hysically-guided **D**ata-space **R**ectified **F**low framework for radar nowcasting, called **PDRF**. PDRF is designed to couple generative sharpness with trajectory stability. First, adapting the manifold insights from generative modeling (Li & He, 2025) to the temporal domain, we introduce Manifold-Anchored Spatiotemporal Dynamics (MASD) via a data-space parameterization, which redefines the network as a manifold projector. By predicting the clean future sequence to analytically induce the vector field, this approach introduces an intrinsic restoring force that explicitly anchors the integration trajectory to valid data states, significantly mitigating numerical drift. Second, to bridge the gap between data-driven generation and physical laws, we inject a soft physics-guided training signal. We utilize Semi-Lagrangian advection as a soft teacher in the velocity domain, forcing the model to respect large-scale transport coherence while retaining the flexibility to learn local nonlinear evolution from data. Third, to instantiate this framework, we design a Motion- and Frequency-Aware Backbone that integrates condition-guided alignment and wavelet-based fusion, ensuring that the architecture is structurally aligned with the spatiotemporal nature of radar echoes.

In summary, our main contributions are as follows:

- We identify velocity-regression error as a primary driver of off-manifold drift and the resulting geometric instability in the conditional variant of Rectified Flow for nowcasting. MASD via data-space parameterization re-casts the generator as a manifold projector, introducing an intrinsic restoring feedback that maintains manifold consistency throughout long-horizon ODE integration.

- We introduce a Physics-Guided Velocity Regularizer that leverages a Semi-Lagrangian advection prior. This mechanism acts as a soft constraint, encouraging globally coherent transport without suppressing the learning of fine-grained intensity evolution.

- We instantiate PDRF with a specialized backbone that incorporates Condition-Guided Alignment and Wavelet-Guided Fusion to address multi-scale misalignment and high-frequency reconstruction challenges in radar data.

## 2. Related Work

**Deterministic Nowcasting: From Extrapolation to Deep Learning.** Classical radar nowcasting relies on optical flow-based extrapolation. Toolkits such as pySTEPS (Pulkkinen et al., 2019) and rainymotion (Ayzel et al., 2019) implement these pipelines, which effectively preserve large-scale transport coherence but fail to model nonlinear intensity growth and decay. To capture complex evolution, deep learning approaches formulate nowcasting as spatiotemporal prediction trained with reconstruction objectives. Early convolutional recurrent architectures, such as ConvLSTM (Shi et al., 2015) and ConvGRU (Ballas et al., 2015; Shi et al., 2017), established foundational baselines for precipitation nowcasting, followed by advanced designs such as MAU (Chang et al., 2021), SimVP (Gao et al., 2022b), and the transformer-based Earthformer (Gao et al., 2022a). Recent works have further integrated physical constraints (PhyDNet (Guen & Thome, 2020)) or frequency-domain disentanglement (AlphaPre (Lin et al., 2025)) to improve fidelity. SSRF-Net (Luo et al., 2025) also attempts to mitigate long-term degradation via a scheduled sliding-window framework. However, despite their ability to model nonlinear dynamics, deterministic methods inherently minimize pixel-wise errors, leading to blurred predictions and underestimated peak intensities as the forecast horizon extends.

**Generative Forecasting and Physics-Aware Designs.** To resolve the over-smoothing issue of deterministic models, probabilistic and generative approaches have been widely adopted to represent uncertainty and recover fine-grained details. GAN-based models like DGMR (Ravuri et al., 2021) and large-capacity systems like NowcastNet (Zhang et al., 2023) generate realistic radar fields by sampling diverse futures. Diffusion models, such as DiffCast (Yu et al., 2024) and physics-driven diffusion networks (Wang et al., 2024), further advance this direction by explicitly modeling stochastic variations. Parallel to pure generation, researchers have sought to inject physical priors—such as advection constraints or physics-informed discriminator supervision—into learning pipelines to retain coherent transport (Yin et al., 2024; Gia et al., 2025). However, a crit-

ical challenge remains: iterative generative sampling often accumulates small inconsistencies over time, leading to trajectory-level drift where the generated weather systems fracture or lose physical coherence over long lead times; recent hybrid/gray-box designs explicitly evolve advection–diffusion-style dynamics to mitigate such drift (Chen et al., 2025b). While PercpCast (Feng et al., 2025) introduces perceptual constraints to a rectified-flow generator, it does not explicitly address the numerical stability of the transport ODE itself.

**Rectified Flow and Data-Space Parameterization.** Rectified Flow (RF) learns a deterministic transport ODE that maps a tractable noise distribution to the data distribution via flow matching (Liu et al., 2022; Lipman et al., 2022). In our setting, we aim to model the distribution of a future sequence $X_1 \in \mathbb{R}^{K_{out} \times H \times W}$. RF samples $X_0 \sim \mathcal{N}(0, I)$ in the same space as $X_1$ and defines the straight-line coupling

$$Z_t = (1-t)X_0 + tX_1, \quad t \in [0,1], \quad (1)$$

with generative dynamics

$$dZ_t = v_\theta(Z_t, t)\, dt. \quad (2)$$

The flow-matching objective regresses the constant path velocity:

$$\mathcal{L}_{\text{RF}} = \mathbb{E}_{t, X_0, X_1}\left[\|v_\theta(Z_t, t) - (X_1 - X_0)\|^2\right]. \quad (3)$$

Standard RF implementations directly predict the flow-space velocity. In high-dimensional spaces, however, regressing this latent derivative is prone to error accumulation. (Li & He, 2025) recently demonstrated that for unconditional image generation, predicting clean data ($x$-prediction) is empirically superior under the manifold assumption. While their work focuses on generation quality in static domains, we identify a distinct theoretical advantage of this parameterization for dynamical systems: it acts as a geometric stabilizer against trajectory drift. In this work, we extend this insight to physical forecasting and, crucially, augment it with a mechanism-aligned Semi-Lagrangian regularizer to ensure that the MASD-anchored trajectory also adheres to kinematic transport laws.

## 3. Methodology

We formulate precipitation nowcasting as a conditional generative modeling task and adopt the conditional variant of Rectified Flow, referred to as Conditional Rectified Flow (CRF), where the dynamics are conditioned on the historical context $C$. Building on CRF, we propose Physically-guided Data-space Rectified Flow (PDRF), which instantiates a data-space parameterization to stabilize long-horizon ODE integration and further incorporates a physics-guided velocity regularizer for transport coherence. Specifically,

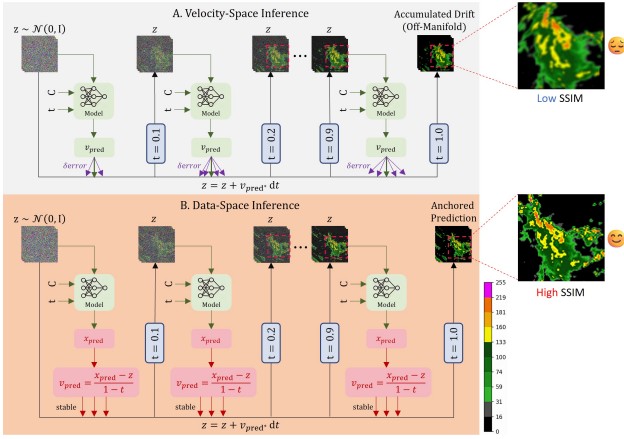

*Figure 2.* **Geometric Stabilization via Parameterization.** (A) Velocity-space inference directly regresses the flow velocity, allowing errors to accumulate during ODE integration and causing off-manifold drift, degraded echo morphology, and lower SSIM. (B) Data-space inference predicts the clean future state $x_{pred}$ and analytically induces the velocity field, producing an implicit restoring effect that anchors the trajectory, mitigates drift accumulation, and improves structural fidelity with higher SSIM.

we introduce Manifold-Anchored Spatiotemporal Dynamics (MASD) via data-space prediction, where the network predicts the clean future target and analytically induces a stabilizing vector field that mitigates off-manifold drift, as illustrated in Fig. 2. To further ensure physically plausible evolution over extended horizons, we incorporate a Semi-Lagrangian guidance signal that regularizes the induced velocity field. Finally, we detail the PDRF architecture, which instantiates these geometric and physical constraints within a unified conditional generative model.

### 3.1. Manifold-Anchored Spatiotemporal Dynamics (MASD) via Data-Space Parameterization

In CRF, the generative dynamics are typically parameterized by directly regressing the instantaneous flow velocity $v_\theta$. While theoretically sound in the continuous-time limit, we posit that this velocity-space parameterization is ill-suited for high-dimensional spatiotemporal forecasting. Under the Manifold Assumption (Chapelle et al., 2006), realistic precipitation sequences concentrate on a low-dimensional physical manifold embedded in the high-dimensional pixel space, shaped by underlying laws (e.g., advection–diffusion). The flow velocity, as a differential quantity linking signal and noise across time, inevitably contains high-frequency, off-manifold components that are chaotic and difficult to model. Consequently, minor velocity estimation errors accumulate during numerical ODE integration over long horizons and can drive the sampling trajectory into invalid regions of the state space—a failure mode we term off-manifold drift.

To structurally enforce physical consistency, we propose

a data-space parameterization that reconfigures the inference dynamics. Instead of modeling the local derivative, our network $f_\theta$ acts as a manifold-attracting denoiser (approximately a projector under the manifold assumption) by predicting a globally consistent clean future state at each timestep:

$$\hat{X}_1 = f_\theta(Z_t, t, C). \tag{4}$$

Crucially, this parameterization induces a velocity field analytically constrained to point toward the predicted physical support:

$$\hat{v}_\theta(Z_t, t, C) = \frac{\hat{X}_1 - Z_t}{\bar{t}}, \qquad \bar{t} = \max(1 - t, \varepsilon). \tag{5}$$

Unlike direct velocity regression, this analytical coupling transforms generation from a pure integration task into an error-correcting feedback process. Specifically, any deviation of $Z_t$ is counteracted by an implicit restoring term whose effective gain scales as $1/\bar{t}$, continuously pulling the trajectory back toward the predicted physical support even under discretization errors. Under the linear coupling $Z_t = (1 - t)X_0 + tX_1$, our velocity target reduces exactly to the standard rectified-flow target when $\bar{t} = 1 - t$.

**Geometric Stabilization via Restoring Force.** Let $\epsilon_t$ denote the perturbation from the ideal trajectory. Under the data-space parameterization in Eq. (5), the error dynamics satisfy

$$\frac{d\epsilon_t}{dt} = -\frac{1}{\bar{t}}(I - \mathbf{J}_x)\epsilon_t, \tag{6}$$

where $\mathbf{J}_x$ is the Jacobian of $f_\theta$ with respect to $Z_t$. As $t \to 1$, the effective gain $1/\bar{t}$ increases sharply (bounded by $\varepsilon$ in practice), which preferentially damps the off-manifold (normal) components of perturbations and thereby suppresses off-manifold drift under discretization errors, improving long-horizon stability. We provide the detailed derivation and stability analysis in Appendix A.

Although inference operates in this geometrically stable data space, we express learning in the induced velocity form for two reasons: (i) it is algebraically consistent with our ODE parameterization, and (ii) it allows the physics-guided regularizer to be imposed in the same domain as the flow-matching objective:

$$\mathcal{L}_{\text{flow}} = \mathbb{E}_{t, X_0, X_1, C}\left[\left\|\hat{v}_\theta(Z_t, t, C) - \frac{X_1 - Z_t}{\bar{t}}\right\|^2\right]. \tag{7}$$

### 3.2. Physics-Guided Velocity Regularization

A fundamental challenge in generative nowcasting is balancing high-frequency realism with low-frequency kinematic consistency. Purely data-driven generators often hallucinate realistic textures that violate physical transport laws (e.g.,

inconsistent motion), while rigid physical constraints can suppress the learning of complex nonlinear evolution. We propose a decoupled regularization strategy: we introduce a physics-guided velocity regularizer $\mathcal{L}_{\text{phy}}$ using a Semi-Lagrangian advection prior. This regularizer constrains the global transport component of the induced vector field, while allowing local intensity evolution (growth, decay, deformation) to be learned from data.

**Semi-Lagrangian prior over the forecast horizon.** Given the conditional context $C = \{X_{\tau - K_{\text{in}} + 1}, \ldots, X_\tau\}$, where $X_\tau$ is the latest observed frame, we construct a Semi-Lagrangian prior sequence

$$X_{\text{SL}} \triangleq \{X_{\text{SL}}^{(1)}, \ldots, X_{\text{SL}}^{(K_{\text{out}})}\} \in \mathbb{R}^{K_{\text{out}} \times H \times W}, \tag{8}$$

spanning the same forecast horizon as the target sequence $X_1$.

We estimate a dense motion field $\mathbf{u} = (u, v)$ (pixel displacement per time step) from the last two historical frames using the Farneback method:

$$\mathbf{u} = \text{Flow}_{\text{FB}}(X_{\tau - 1}, X_\tau), \tag{9}$$

where inputs are normalized to $[0, 1]$ prior to optical flow estimation. Here, $\text{Flow}_{\text{FB}}(\cdot, \cdot)$ denotes the Farneback dense optical-flow operator that returns a per-pixel 2D displacement field $\mathbf{u} \in \mathbb{R}^{2 \times H \times W}$, where $u(x, y)$ and $v(x, y)$ are the horizontal (right-positive) and vertical (down-positive) displacements, respectively, measured in pixels per time step. In implementation, the normalized frames are converted to a fixed uint8 scale via $X \mapsto \lfloor 255 \text{clip}(X, 0, 1) \rfloor$ before applying Farneback, avoiding per-frame min–max rescaling. For numerical robustness, the estimated flow is Gaussian-smoothed with $\sigma = 1.2$ and clipped to a bounded displacement magnitude proportional to $\min(H, W)$.

For each lead time $k \in \{1, \ldots, K_{\text{out}}\}$, Semi-Lagrangian backtracing defines the upstream coordinates as

$$(x_s, y_s) = (x - k \cdot u(x, y), \; y - k \cdot v(x, y)). \tag{10}$$

The prior frame is then obtained by warping the latest observation:

$$X_{\text{SL}}^{(k)} = \mathcal{W}_k(X_\tau; \mathbf{u}). \tag{11}$$

Here, $\mathcal{W}_k(\cdot; \mathbf{u})$ is the $k$-step Semi-Lagrangian warping operator induced by backtracing, defined pointwise as

$$\left[\mathcal{W}_k(X; \mathbf{u})\right](x, y) = X(y_s, x_s),$$

where $(x_s, y_s)$ are generally non-integer coordinates and are evaluated by interpolation. In implementation, $\mathcal{W}_k$ uses cubic interpolation (order 3) with nearest boundary handling, and the coordinates are clipped to the valid image domain to prevent out-of-range sampling.

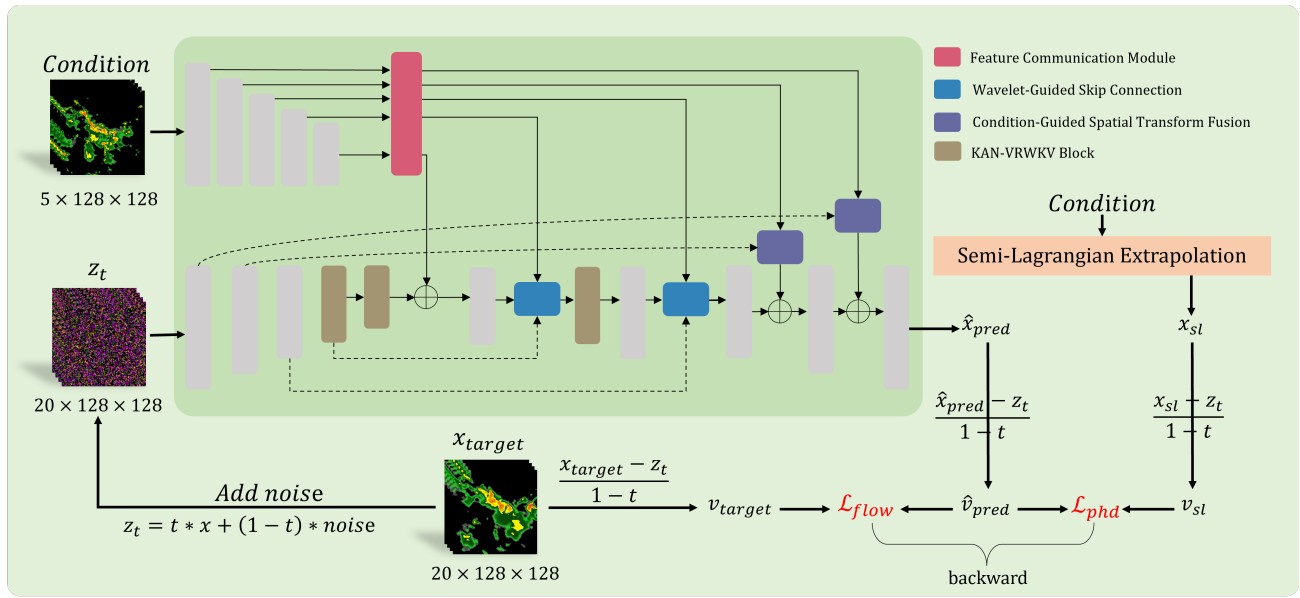

*Figure 3.* **Overview of the proposed Physically-guided Data-space Rectified Flow framework.** The model conditions on historical radar echoes and uses the PDRF backbone to predict the clean future sequence $\hat{X}_1$. The architecture integrates KAN-VRWKV blocks for spatiotemporal modeling, the Feature Communication Module (FCM) for multi-scale interaction, Wavelet-Guided Skip Connections (WGSC) for frequency-aware fusion, and Condition-Guided Spatial Transform Fusion (CGSTF) for feature alignment. The training objective combines a data-driven flow-matching loss with a physics-guided velocity regularizer, where a Semi-Lagrangian advection prior is mapped into the induced velocity space.

**Mapping the physics prior into CRF velocity space.** During training, CRF constructs the intermediate state $Z_t = tX_1 + (1 - t)X_0$ at continuous flow time $t \in [0, 1]$, with $X_0 \sim \mathcal{N}(0, I)$. We treat $X_{\mathrm{SL}}$ as a physics-consistent surrogate for $X_1$ and map it into the CRF velocity space using the same stabilized scaling as in Eq. (5):

$$v_{\mathrm{SL}} = \frac{X_{\mathrm{SL}} - Z_t}{\bar{t}}. \tag{12}$$

Here, $\bar{t}$ denotes the stabilized time scaling used in Eq. (5) to avoid numerical instability for small $t$. The resulting $v_{\mathrm{SL}}$ acts as a proxy velocity induced by the physics prior in the CRF state space, distinct from the physical motion field $\mathbf{u}$.

**Physics-guided regularization.** We use the advection-derived velocity $v_{\mathrm{SL}}$ as a soft target for the physics-guided velocity regularizer in the induced velocity domain. The regularization term is defined as:

$$\mathcal{L}_{\mathrm{phy}} = \mathbb{E}_{t, X_0, X_1, C} \left[ \|\hat{v}_\theta(Z_t, t, C) - v_{\mathrm{SL}}\|^2 \right]. \tag{13}$$

Crucially, this term does not force the prediction to be identical to the advection result (which would result in blurry forecasts). Instead, we treat $X_{\mathrm{SL}}$ as a coarse, transport-consistent proxy of $X_1$ and use it only as a soft target in the induced velocity domain. When optimized alongside the data-driven loss $\mathcal{L}_{\mathrm{flow}}$, $\mathcal{L}_{\mathrm{phy}}$ acts as a kinematic scaffold: it discourages large-scale velocity directions that deviate

drastically from advection trends, while still allowing the model to learn local growth/decay/deformation beyond the capacity of simple advection.

The overall training objective is

$$\mathcal{L}_{\mathrm{total}} = \mathcal{L}_{\mathrm{flow}} + \lambda_{\mathrm{phy}} \mathcal{L}_{\mathrm{phy}}. \tag{14}$$

### 3.3. Implementation: The PDRF Architecture

The overall architecture of the proposed PDRF is illustrated in Fig. 3. The conditional denoiser is parameterized by a U-shaped hierarchical backbone tailored for spatiotemporal nowcasting. At shallow stages, we address advection-induced misalignment using a Condition-Guided Spatial Transform Fusion module, which estimates a dense displacement field from the conditional radar echoes and applies differentiable warping to align backbone features with the conditional context before fusion. To enable effective multi-scale representation learning, we integrate a Feature Communication Module on the conditional-encoder outputs; it propagates information bidirectionally across pyramid scales so that local features are modulated by global context via pixel-wise adaptive selection. In the deep bottleneck, we insert KAN-VRWKV blocks to capture long-range temporal dependencies with linear-time global spatiotemporal context modeling; each block contains a VRWKV temporal-mixing unit coupled with a Kolmogorov–Arnold Network. Finally, we employ Wavelet-Guided Skip Connections: conditional

*Table 1.* Experiment results on four radar datasets. The best results are highlighted in **bold** and the second-best results are underlined.

| Method | Params(M)↓ | GFLOPs↓ | SEVIR | | | | | | MeteoNet | | | | | |
|---|---|---|---|---|---|---|---|---|---|---|---|---|---|---|
| | | | CSI-M↑ | CSI-181↑ | CSI-219↑ | HSS↑ | SSIM↑ | MSE↓ | CSI-M↑ | CSI-24↑ | CSI-32↑ | HSS↑ | SSIM↑ | MSE↓ |
| pySTEPS | – | – | 0.2830 | 0.1266 | 0.0708 | 0.3673 | 0.6654 | 652.83 | 0.3647 | 0.3552 | 0.2273 | 0.4964 | 0.8118 | 16.42 |
| ConvGRU | 5.990 | 1962.556 | 0.2951 | 0.0846 | 0.0412 | 0.3696 | 0.6015 | 369.72 | 0.3463 | 0.2911 | 0.1495 | 0.4741 | 0.7719 | 13.67 |
| MAU | 51.218 | 291.462 | 0.3029 | 0.1129 | 0.0487 | 0.3799 | 0.6647 | 354.01 | 0.3162 | 0.2896 | 0.0920 | 0.4389 | 0.8029 | 11.88 |
| SimVP | 11.038 | 51.089 | 0.3172 | 0.1061 | 0.0579 | 0.3988 | 0.6419 | 382.44 | 0.3439 | 0.3078 | 0.1072 | 0.4516 | 0.7921 | 14.82 |
| FourCastNet | 57.014 | 58.372 | 0.2621 | 0.0788 | 0.0287 | 0.3426 | 0.5849 | 411.63 | 0.2969 | 0.2601 | 0.1146 | 0.4282 | 0.6324 | 14.21 |
| Earthformer | 60.246 | 849.376 | 0.2837 | 0.0915 | 0.0211 | 0.3608 | 0.6786 | 361.52 | 0.3276 | 0.2810 | 0.1299 | 0.4579 | 0.7648 | 13.06 |
| PhyDNet | 5.829 | 286.214 | 0.3098 | 0.0987 | 0.0331 | 0.3749 | 0.6474 | 356.02 | 0.3316 | 0.3269 | 0.1309 | 0.4604 | 0.7954 | 15.96 |
| EarthFarseer | 94.093 | 907.237 | 0.2949 | 0.1058 | 0.0381 | 0.3907 | 0.6218 | 389.88 | 0.3488 | 0.3093 | 0.1429 | 0.4802 | 0.7425 | 12.69 |
| NowcastNet | 33.876 | 17.634 | 0.2868 | 0.0712 | 0.0397 | 0.3461 | **0.6950** | 411.20 | 0.3360 | 0.3279 | 0.1669 | 0.4834 | 0.7757 | 14.07 |
| DiffCast | 52.079 | 30616.382 | 0.3050 | 0.1300 | 0.0582 | 0.3996 | 0.6482 | 559.59 | 0.3512 | 0.3340 | 0.1808 | 0.4846 | 0.7887 | 17.93 |
| AlphaPre | 89.011 | 1550.903 | 0.3259 | 0.1332 | 0.0545 | 0.4110 | 0.6884 | **345.18** | 0.3824 | 0.3633 | 0.2002 | 0.5164 | 0.7968 | 12.74 |
| PercpCast | 55.874 | 324.926 | 0.3492 | 0.1928 | 0.1022 | 0.4538 | 0.6852 | 449.65 | 0.4038 | 0.3932 | 0.2206 | 0.5436 | 0.8266 | 12.96 |
| PDRF (Ours) | 27.153 | 28.633 | **0.3622** | **0.2110** | **0.1199** | **0.4678** | 0.6869 | 451.80 | **0.4386** | **0.4288** | **0.2684** | **0.5747** | **0.8385** | **11.14** |
| Method | Params(M)↓ | GFLOPs↓ | Shanghai | | | | | | CIKM | | | | | |
| | | | CSI-M↑ | CSI-35↑ | CSI-40↑ | HSS↑ | SSIM↑ | MSE↓ | CSI-M↑ | CSI-35↑ | CSI-40↑ | HSS↑ | SSIM↑ | MSE↓ |
| pySTEPS | – | – | 0.3719 | 0.3376 | 0.2514 | 0.4995 | 0.7877 | 42.11 | 0.2856 | 0.2141 | 0.1550 | 0.3779 | 0.5427 | 80.83 |
| ConvGRU | 5.990 | 1962.556 | 0.3687 | 0.3096 | 0.2133 | 0.4950 | 0.7928 | 32.17 | 0.3166 | 0.1942 | 0.1322 | 0.3927 | 0.6635 | 38.94 |
| MAU | 51.218 | 291.462 | 0.3904 | 0.3698 | 0.2350 | 0.5289 | 0.7066 | 31.92 | 0.2976 | 0.2120 | 0.1185 | 0.3991 | 0.6206 | 39.65 |
| SimVP | 11.038 | 51.089 | 0.3922 | 0.3471 | 0.2448 | 0.5267 | 0.7618 | 35.86 | 0.3139 | 0.1972 | 0.1388 | 0.4046 | **0.6679** | 36.75 |
| FourCastNet | 57.014 | 58.372 | 0.3506 | 0.3189 | 0.2001 | 0.4795 | 0.5714 | 31.05 | 0.2905 | 0.1926 | 0.0959 | 0.3874 | 0.4483 | 37.56 |
| Earthformer | 60.246 | 849.376 | 0.3589 | 0.3102 | 0.1957 | 0.4928 | 0.7159 | 33.94 | 0.3001 | 0.2118 | 0.1308 | 0.4078 | 0.6149 | 36.07 |
| PhyDNet | 5.829 | 286.214 | 0.3599 | 0.3319 | 0.2108 | 0.4880 | 0.7865 | 37.62 | 0.3109 | 0.1989 | 0.1343 | 0.3864 | 0.6422 | 40.93 |
| EarthFarseer | 94.093 | 907.237 | 0.4011 | 0.3534 | 0.2420 | 0.5409 | 0.5269 | 34.23 | 0.2983 | 0.2112 | 0.1317 | 0.3984 | 0.6227 | 38.22 |
| NowcastNet | 33.876 | 17.634 | 0.3886 | 0.3681 | 0.2389 | 0.5405 | 0.7764 | 34.77 | 0.3074 | 0.1879 | 0.1269 | 0.3939 | 0.6537 | 39.51 |
| DiffCast | 52.079 | 30616.382 | 0.4089 | 0.3740 | 0.2606 | 0.5476 | 0.7879 | 36.35 | 0.3159 | 0.2009 | 0.1457 | 0.4085 | 0.6499 | 42.78 |
| AlphaPre | 89.011 | 1550.903 | 0.4178 | 0.3854 | 0.2615 | 0.5534 | 0.7951 | **28.02** | 0.3194 | 0.2068 | 0.1416 | 0.4137 | 0.6568 | **35.18** |
| PercpCast | 55.874 | 324.926 | 0.3995 | 0.3628 | 0.2577 | 0.5403 | 0.7941 | 39.91 | 0.3162 | 0.2223 | 0.1553 | 0.4107 | 0.6431 | 45.50 |
| PDRF (Ours) | 27.153 | 28.633 | **0.4409** | **0.4066** | **0.3032** | **0.5772** | **0.7983** | 30.02 | **0.3329** | **0.2380** | **0.1720** | **0.4312** | 0.6619 | 37.69 |

Note: "Params (M)" and "GFLOPs" were computed assuming an input of $(1, 5, 1, 128, 128)$ and an output of $(1, 20, 1, 128, 128)$.

features are decomposed into frequency sub-bands via discrete wavelet transforms to form frequency-aware gates that selectively modulate fusion, preserving high-frequency echo structure while suppressing noise.

# 4. Experiments

## 4.1. Experimental Setting

**Datasets.** We evaluate PDRF on four radar nowcasting benchmarks. SEVIR (Veillette et al., 2020) contains 20,393 storm events over the Continental United States from 2017–2020, with a 5-minute cadence and a $384 \times 384$ km spatial footprint. We use the VIL product with pixel values ranging from 0 to 255 and thresholds $\{16, 74, 133, 160, 181, 219\}$. MeteoNet (Larvor et al., 2020) provides radar data over northwestern France from 2016–2018, covering a $550 \times 550$ km area at a 5-minute cadence, with reflectivity values of 0–70 dBZ and thresholds $\{12, 18, 24, 32\}$ dBZ. Shanghai Radar (Chen et al., 2020) contains radar echoes collected over Shanghai, China, from October 2015 to July 2018, covering a $501 \times 501$ km region at an approximately 6-minute cadence, with reflectivity values of 0–70 dBZ and thresholds $\{20, 30, 35, 40\}$ dBZ. CIKM 2017 AnalytiCup (Yao & Li, 2017) covers a $101 \times 101$ km area over Guangdong, China, with 6-minute radar frames, reflectivity values of 0–76 dBZ, and thresholds $\{20, 30, 35, 40\}$ dBZ. For all datasets, we use chronological train/validation/test splits to avoid temporal leakage and resize all radar frames to $128 \times 128$ pixels.

Unless otherwise specified, models predict the next $K = 20$ frames from the previous $J = 5$ frames, while CIKM uses a $5 \rightarrow 10$ setting corresponding to a 60-minute forecast horizon.

**Metrics.** We evaluate nowcasting quality with four widely used metrics, including the mean Critical Success Index (CSI-M), Heidke Skill Score (HSS), Structural Similarity Index Measure (SSIM), and mean squared error (MSE), consistent with common practice in radar precipitation forecasting (Lin et al., 2025). CSI assesses event-detection performance by jointly considering missed events and false alarms. We report CSI under multiple intensity thresholds and further summarize them with CSI-M to provide an overall measure of detection accuracy. HSS reflects the skill of the forecasts relative to chance. In addition, SSIM measures structural consistency between predicted and target fields, whereas MSE captures the average pixel-wise deviation in intensity.

**Implementation Details.** All models were trained for 300 epochs using the AdamW optimizer with an initial learning rate of $2 \times 10^{-4}$ and a cosine annealing schedule. We used a batch size of 8 and maintained an exponential moving average (EMA) of model weights with a decay rate of 0.95. For preprocessing, all input frames were resized to $128 \times 128$ and linearly normalized to the $[0, 1]$ range according to each dataset's native value scale. We selected the final checkpoint based on the best validation CSI-M. During inference, forecasts were generated with a fixed 5-step ODE

sampler.

**Baselines.** We benchmark our method against twelve representative baselines categorized into four groups: (1) Optical flow extrapolation, represented by the non-learning toolkit pySTEPS (Pulkkinen et al., 2019); (2) Deterministic deep predictors, ranging from recurrent and convolutional units (ConvGRU (Shi et al., 2017), MAU (Chang et al., 2021), SimVP (Gao et al., 2022b)) to Transformer and Fourier-based operators (FourCastNet (Pathak et al., 2022), Earthformer (Gao et al., 2022a), EarthFarseer (Wu et al., 2024)); (3) Physics-informed factorized models, including PhyDNet (Guen & Thome, 2020) and AlphaPre (Lin et al., 2025); and (4) Generative models, namely the stochastic NowcastNet (Zhang et al., 2023), rectified-flow-based PercpCast (Feng et al., 2025), and diffusion-based DiffCast (Yu et al., 2024). For the ablation studies, the base backbone architecture is instantiated using U-KAN (Li et al., 2025a).

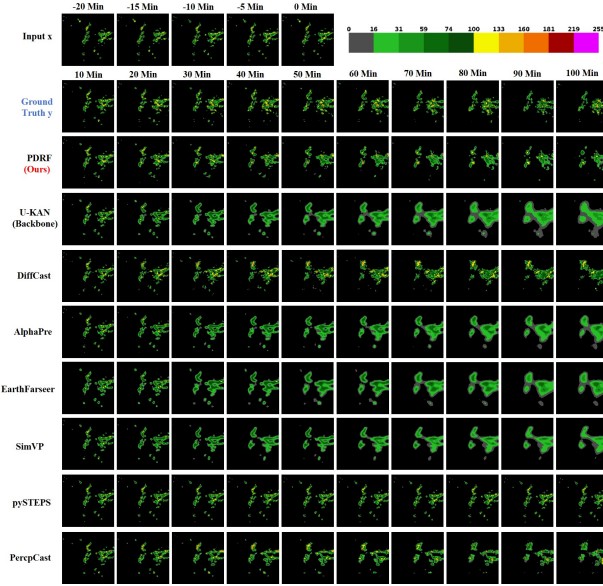

*Figure 4.* Qualitative comparison on the SEVIR benchmark.

## 4.2. Experimental Results

Table 1 summarizes the quantitative comparison on four radar nowcasting benchmarks. The results show that PDRF effectively alleviates the tension between deterministic over-smoothing and structural instability in generative forecasting, achieving consistently strong performance across the four benchmarks. In terms of over-smoothing mitigation, PDRF performs particularly well in detecting intense precipitation events. It outperforms not only deterministic predictors such as Earthformer and SimVP, but also advanced methods with physical or frequency-aware designs, such as AlphaPre. This indicates that the proposed framework can better characterize the uncertainty and multi-modal evolution of intense radar echoes, thereby reducing the regression-

to-the-mean tendency caused by reconstruction-oriented objectives and preserving high-intensity echo structures more effectively. In terms of generative stability, PDRF preserves stronger structural fidelity than representative generative models such as PercpCast, as reflected by its leading SSIM results on MeteoNet and Shanghai. This advantage is also supported by Fig. 4, where competing methods tend to over-smooth radar echoes or produce fragmented textures at later lead times. In contrast, PDRF maintains sharper precipitation boundaries, more coherent transport patterns, and a more faithful intensity distribution. These observations support the claim that the data-space parameterization helps anchor the generation process to the valid radar data manifold. Finally, the long-range evaluation in Fig. 5 further demonstrates the robustness of PDRF. The model consistently maintains an advantage across different lead-time intervals and preserves a stable margin over competing methods throughout the forecast horizon. The increasing gap over pySTEPS at later steps also suggests that PDRF learns nonlinear precipitation evolution beyond simple advection, while remaining stable during long-range extrapolation.

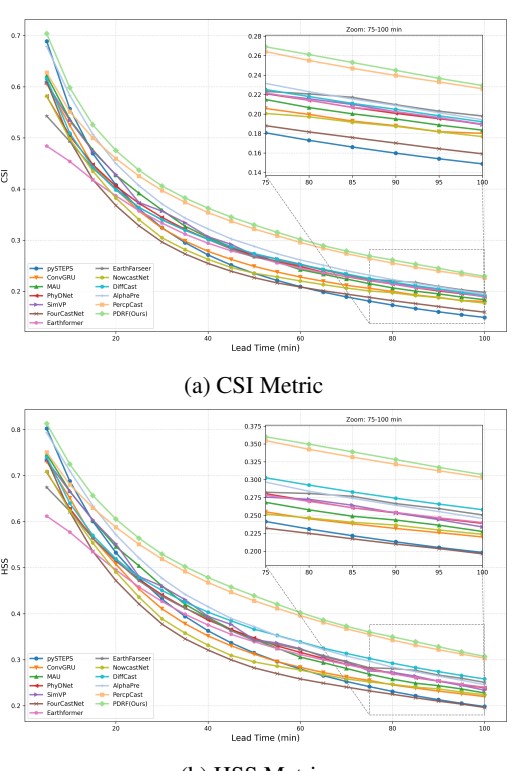

(a) CSI Metric

(b) HSS Metric

*Figure 5.* CSI and HSS metrics at different prediction time steps for various methods on the SEVIR dataset.

## 4.3. Ablation Studies

**Contribution of Framework Components.** Table 2 presents a progressive ablation study across four datasets

*Table 2.* Ablation study results on four datasets. We progressively add each component to demonstrate their individual contributions. The best results are shown in **bold**. The value shown at the lower-right corner of each entry indicates the relative change with respect to the previous step, i.e., relative improvement or relative degradation.

| Method | SEVIR | | | | | | MeteoNet | | | | | |
|---|---|---|---|---|---|---|---|---|---|---|---|---|
| | CSI-M↑ | CSI-181↑ | CSI-219↑ | HSS↑ | SSIM↑ | MSE↓ | CSI-M↑ | CSI-24↑ | CSI-32↑ | HSS↑ | SSIM↑ | MSE↓ |
| Backbone | 0.3095 | 0.1049 | 0.0401 | 0.3875 | 0.6329 | 461.26 | 0.3793 | 0.3453 | 0.1601 | 0.5026 | 0.8059 | 13.31 |
| + CRF | $0.3456_{+11.66\%}$ | $0.1823_{+73.78\%}$ | $0.0912_{+127.43\%}$ | $0.4234_{+9.26\%}$ | $0.6155_{-2.75\%}$ | $451.23_{+2.17\%}$ | $0.4021_{+6.01\%}$ | $0.3834_{+11.03\%}$ | $0.2298_{+43.54\%}$ | $0.5387_{+7.18\%}$ | $0.6812_{-15.47\%}$ | $12.67_{+4.81\%}$ |
| + CGSTF | $0.3389_{-1.94\%}$ | $0.1887_{+3.51\%}$ | $0.0945_{+3.62\%}$ | $0.4312_{+1.84\%}$ | $0.6182_{+0.44\%}$ | $448.91_{+0.51\%}$ | $0.4156_{+3.36\%}$ | $0.3956_{+3.18\%}$ | $0.2401_{+4.48\%}$ | $0.5456_{+1.28\%}$ | $0.6854_{+0.62\%}$ | $12.34_{+2.61\%}$ |
| + WGSC | $0.3478_{+2.63\%}$ | $0.1834_{-2.81\%}$ | $0.0978_{+3.49\%}$ | $0.4287_{-0.58\%}$ | $0.6205_{+0.37\%}$ | $442.16_{+1.50\%}$ | $0.4087_{-1.66\%}$ | $0.3912_{-1.11\%}$ | $0.2356_{-1.87\%}$ | $0.5423_{-0.61\%}$ | $0.6898_{+0.64\%}$ | $12.18_{+1.30\%}$ |
| + FCM | $0.3521_{+1.24\%}$ | $0.1945_{+6.05\%}$ | $0.1023_{+4.60\%}$ | $0.4456_{+3.93\%}$ | $0.6228_{+0.37\%}$ | $439.87_{+0.52\%}$ | $0.4198_{+2.72\%}$ | $0.4034_{+3.12\%}$ | $0.2445_{+3.78\%}$ | $0.5512_{+1.64\%}$ | $0.6923_{+0.36\%}$ | $12.03_{+1.23\%}$ |
| + VRWKV | $0.3552_{+0.88\%}$ | $0.1967_{+1.13\%}$ | $0.1070_{+4.59\%}$ | $0.4576_{+2.69\%}$ | $0.6241_{+0.21\%}$ | $435.54_{+0.98\%}$ | $0.4213_{+0.36\%}$ | $0.4087_{+1.31\%}$ | $0.2477_{+1.31\%}$ | $0.5563_{+0.93\%}$ | $0.6956_{+0.48\%}$ | $11.91_{+1.00\%}$ |
| + Data-Pred | $0.3522_{-0.84\%}$ | $0.1965_{-0.10\%}$ | $0.1077_{+0.65\%}$ | $0.4547_{-0.63\%}$ | $0.6529_{+4.61\%}$ | $441.34_{-1.33\%}$ | $0.4289_{+1.80\%}$ | $0.4188_{+2.47\%}$ | $0.2576_{+4.00\%}$ | $0.5627_{+1.15\%}$ | $0.8348_{+19.99\%}$ | $11.80_{+0.92\%}$ |
| + $\mathcal{L}_{phys}$ | $\mathbf{0.3622}_{+2.84\%}$ | $\mathbf{0.2110}_{+7.38\%}$ | $\mathbf{0.1199}_{+11.33\%}$ | $\mathbf{0.4678}_{+2.88\%}$ | $\mathbf{0.6869}_{+5.21\%}$ | $451.80_{-2.37\%}$ | $\mathbf{0.4386}_{+2.26\%}$ | $\mathbf{0.4288}_{+2.39\%}$ | $\mathbf{0.2684}_{+4.19\%}$ | $\mathbf{0.5747}_{+2.13\%}$ | $\mathbf{0.8385}_{+0.44\%}$ | $\mathbf{11.14}_{+5.59\%}$ |

| Method | Shanghai | | | | | | CIKM | | | | | |
|---|---|---|---|---|---|---|---|---|---|---|---|---|
| | CSI-M↑ | CSI-35↑ | CSI-40↑ | HSS↑ | SSIM↑ | MSE↓ | CSI-M↑ | CSI-35↑ | CSI-40↑ | HSS↑ | SSIM↑ | MSE↓ |
| Backbone | 0.4002 | 0.3550 | 0.2420 | 0.5260 | 0.6583 | 34.01 | 0.2988 | 0.2023 | 0.1276 | 0.3849 | 0.6043 | 46.99 |
| + CRF | $0.4087_{+2.12\%}$ | $0.3734_{+5.18\%}$ | $0.2687_{+11.03\%}$ | $0.5345_{+1.62\%}$ | $0.6420_{-2.48\%}$ | $34.78_{-2.26\%}$ | $0.3098_{+3.68\%}$ | $0.2134_{+5.49\%}$ | $0.1389_{+8.86\%}$ | $0.3912_{+1.64\%}$ | $0.5423_{-10.27\%}$ | $46.23_{+1.62\%}$ |
| + CGSTF | $0.4023_{-1.57\%}$ | $0.3698_{-0.96\%}$ | $0.2612_{-2.79\%}$ | $0.5289_{-1.05\%}$ | $0.6510_{+1.40\%}$ | $34.92_{-0.40\%}$ | $0.3067_{-1.00\%}$ | $0.2098_{-1.69\%}$ | $0.1356_{-2.38\%}$ | $0.3876_{-0.92\%}$ | $0.5467_{+0.81\%}$ | $45.87_{+0.78\%}$ |
| + WGSC | $0.4156_{+3.31\%}$ | $0.3821_{+3.33\%}$ | $0.2756_{+5.51\%}$ | $0.5467_{+3.37\%}$ | $0.6605_{+1.46\%}$ | $33.89_{+2.95\%}$ | $0.3156_{+2.90\%}$ | $0.2198_{+4.77\%}$ | $0.1434_{+5.75\%}$ | $0.4012_{+3.51\%}$ | $0.5501_{+0.62\%}$ | $45.12_{+1.64\%}$ |
| + FCM | $0.4134_{-0.53\%}$ | $0.3789_{-0.84\%}$ | $0.2734_{-0.80\%}$ | $0.5423_{-0.80\%}$ | $0.6689_{+1.27\%}$ | $33.65_{+0.71\%}$ | $0.3234_{+2.47\%}$ | $0.2334_{+6.19\%}$ | $0.1567_{+9.28\%}$ | $0.4187_{+4.36\%}$ | $0.5534_{+0.60\%}$ | $44.34_{+1.73\%}$ |
| + VRWKV | $0.4198_{+1.55\%}$ | $0.3867_{+2.06\%}$ | $0.2823_{+3.25\%}$ | $0.5583_{+2.95\%}$ | $0.6751_{+0.93\%}$ | $32.91_{+2.20\%}$ | $0.3292_{+1.79\%}$ | $\mathbf{0.2406}_{+3.08\%}$ | $0.1666_{+6.32\%}$ | $0.4278_{+2.17\%}$ | $0.5561_{+0.49\%}$ | $43.16_{+2.66\%}$ |
| + Data-Pred | $0.4348_{+3.58\%}$ | $0.4017_{+3.88\%}$ | $0.3010_{+6.62\%}$ | $0.5710_{+2.28\%}$ | $0.7682_{+13.79\%}$ | $33.03_{-0.36\%}$ | $0.3315_{+0.70\%}$ | $0.2358_{-1.99\%}$ | $0.1640_{-1.56\%}$ | $0.4266_{-0.28\%}$ | $0.6576_{+18.25\%}$ | $42.61_{+1.27\%}$ |
| + $\mathcal{L}_{phys}$ | $\mathbf{0.4409}_{+1.40\%}$ | $\mathbf{0.4066}_{+1.22\%}$ | $\mathbf{0.3032}_{+0.73\%}$ | $\mathbf{0.5772}_{+1.09\%}$ | $\mathbf{0.7983}_{+3.92\%}$ | $\mathbf{30.02}_{+9.13\%}$ | $\mathbf{0.3329}_{+0.42\%}$ | $0.2380_{+0.93\%}$ | $\mathbf{0.1720}_{+4.88\%}$ | $\mathbf{0.4312}_{+1.08\%}$ | $\mathbf{0.6619}_{+0.66\%}$ | $\mathbf{37.69}_{+11.55\%}$ |

to quantify the contribution of each component. For clarity, CRF corresponds to velocity-space prediction (v-pred), whereas the proposed data-space parameterization corresponds to data-space prediction (x-pred). First, integrating CRF training into the backbone yields consistent gains in CSI-M and HSS, indicating that flow matching improves precipitation occurrence modeling. However, this improvement comes at the cost of structural fidelity, as evidenced by a 15.47% drop in SSIM on MeteoNet, suggesting that velocity-space prediction and ODE integration may introduce structural distortions. Second, incorporating CGSTF, WGSC, and FCM further improves CSI and HSS, particularly at higher thresholds such as CSI-40, validating their roles in mitigating spatial misalignment and preserving fine-scale echo details. We then insert VRWKV blocks in series at the deep stages of the KAN backbone to form the KAN-VRWKV module, thereby assessing the contribution of VRWKV-based temporal mixing under this coupled design. Crucially, switching from v-pred to x-pred substantially restores SSIM, yielding a 19.99% increase on MeteoNet without compromising event-based skill. This supports the claim that directly predicting the target sequence in data space mitigates the manifold inconsistency inherent to latent velocity regression. Finally, $\mathcal{L}_{phys}$ consistently improves key metrics, indicating that the Semi-Lagrangian prior provides effective motion scaffolding for coherent transport.

**Impact of Loss and Parameterization.** To examine the physical guidance design and the effect of parameterization, we conduct a multi-dimensional ablation on MeteoNet (Table 3) covering the physical weight $\lambda$, the prediction target, and the loss calculation space. Varying $\lambda \in \{0.0, 0.1, 0.3, 0.5, 0.8, 1.0\}$ shows that a moderate weight ($\lambda = 0.1$) achieves the best overall balance, improving CSI-M by 2.26% over $\lambda = 0.0$, whereas larger weights ($\lambda \geq 0.5$) over-constrain the model and consistently reduce CSI/HSS. Meanwhile, $\lambda = 0.3$ attains the lowest

*Table 3.* Ablation study on the physical regularization weight ($\lambda$), prediction parameterization, and loss calculation space on the MeteoNet dataset. The best results are highlighted in **bold**.

| Method | CSI-M↑ | CSI-24↑ | CSI-32↑ | HSS↑ | SSIM↑ | MSE↓ |
|---|---|---|---|---|---|---|
| *Impact of Physical Regularization Weight ($\lambda$)* | | | | | | |
| $\lambda = 0.0$ | 0.4289 | 0.4188 | 0.2576 | 0.5627 | 0.8348 | 11.80 |
| $\lambda = 0.1$ | **0.4386** | **0.4288** | **0.2684** | **0.5747** | **0.8385** | 11.14 |
| $\lambda = 0.3$ | 0.4292 | 0.4161 | 0.2627 | 0.5646 | 0.8371 | **11.06** |
| $\lambda = 0.5$ | 0.4235 | 0.4100 | 0.2599 | 0.5600 | 0.8346 | 11.36 |
| $\lambda = 0.8$ | 0.4138 | 0.3984 | 0.2523 | 0.5481 | 0.8370 | 11.37 |
| $\lambda = 1.0$ | 0.4129 | 0.3988 | 0.2526 | 0.5484 | 0.8276 | 11.94 |
| *Effect of Parameterization ($\lambda = 0.1$)* | | | | | | |
| v-pred | 0.4115 | 0.3989 | 0.2319 | 0.5456 | 0.7908 | 11.94 |
| x-pred | **0.4386** | **0.4288** | **0.2684** | **0.5747** | **0.8385** | **11.14** |
| *Effect of Loss Calculation Space ($\lambda = 0.1$)* | | | | | | |
| $x$-Space Loss | 0.4281 | 0.4185 | 0.2529 | 0.5618 | 0.8372 | 11.18 |
| $v$-Space Loss | **0.4386** | **0.4288** | **0.2684** | **0.5747** | **0.8385** | **11.14** |

MSE, indicating a slightly stronger emphasis on pixel-wise fidelity at the expense of event-based scores. With $\lambda$ fixed to 0.1, x-pred substantially outperforms v-pred across all metrics: beyond the 6.03% gain in SSIM, it also yields consistent improvements in CSI/HSS, indicating that predicting clean future states provides a more stable learning target and is less sensitive to long-horizon error accumulation than direct velocity regression. Finally, replacing the pixel-space constraint with our velocity-domain formulation ($v$-Space Loss) leads to uniform gains over the $x$-Space Loss, suggesting that expressing both the physical prior and the flow-matching objective in the same derivative space produces better-aligned gradients and thus a more effective optimization signal.

**Rollout Drift Diagnostic.** To directly examine drift accumulation during ODE rollout, we compare v-pred and x-pred under the same architecture, training protocol, and fixed 5-step ODE sampler. To isolate the effect of param-

*Table 4.* Direct diagnostic of rollout drift accumulation. $D_\perp(t)$ denotes the normalized off-path deviation from the ideal rectified-flow path at rollout time $t$, and MeanOffPath averages the deviation over all recorded steps. Lower values are better. Relative Reduction (Rel. Red.) denotes the percentage reduction of x-pred relative to v-pred.

| Dataset | Method | $D_\perp(0.2)$ | $D_\perp(0.4)$ | $D_\perp(0.6)$ | $D_\perp(0.8)$ | $D_\perp(1.0)$ | **MeanOffPath** |
|---|---|---|---|---|---|---|---|
| MeteoNet | v-pred | $0.007119 \pm 0.001861$ | $0.014080 \pm 0.003764$ | $0.021007 \pm 0.005735$ | $0.028247 \pm 0.007877$ | $0.036962 \pm 0.010417$ | $0.021483 \pm 0.005927$ |
| | x-pred | $0.006665 \pm 0.001984$ | $0.013345 \pm 0.003981$ | $0.020085 \pm 0.005998$ | $0.027171 \pm 0.008105$ | $0.035384 \pm 0.010429$ | $0.020530 \pm 0.006097$ |
| | Rel.Red.(%) | 6.38% ↓ | 5.22% ↓ | 4.39% ↓ | 3.81% ↓ | 4.27% ↓ | 4.44% ↓ |
| Shanghai | v-pred | $0.013096 \pm 0.002320$ | $0.025289 \pm 0.004802$ | $0.037079 \pm 0.007597$ | $0.049083 \pm 0.010818$ | $0.062470 \pm 0.014527$ | $0.037403 \pm 0.008002$ |
| | x-pred | $0.010592 \pm 0.002688$ | $0.021334 \pm 0.005393$ | $0.032467 \pm 0.008174$ | $0.044560 \pm 0.011071$ | $0.058547 \pm 0.014170$ | $0.033500 \pm 0.008287$ |
| | Rel.Red.(%) | 19.13% ↓ | 15.64% ↓ | 12.44% ↓ | 9.21% ↓ | 6.28% ↓ | 10.44% ↓ |
| SEVIR | v-pred | $0.013073 \pm 0.006410$ | $0.026146 \pm 0.013141$ | $0.039884 \pm 0.020418$ | $0.054967 \pm 0.030096$ | $0.071360 \pm 0.039223$ | $0.041086 \pm 0.021193$ |
| | x-pred | $0.012761 \pm 0.006953$ | $0.025700 \pm 0.014084$ | $0.039505 \pm 0.021750$ | $0.054568 \pm 0.028481$ | $0.070744 \pm 0.037553$ | $0.040656 \pm 0.022413$ |
| | Rel.Red.(%) | 2.39% ↓ | 1.70% ↓ | 0.95% ↓ | 0.73% ↓ | 0.87% ↓ | 1.05% ↓ |
| CIKM | v-pred | $0.015148 \pm 0.003322$ | $0.030151 \pm 0.006681$ | $0.045355 \pm 0.010110$ | $0.061296 \pm 0.013650$ | $0.079080 \pm 0.017335$ | $0.046206 \pm 0.010203$ |
| | x-pred | $0.014436 \pm 0.003229$ | $0.028904 \pm 0.006463$ | $0.043434 \pm 0.009708$ | $0.058246 \pm 0.012955$ | $0.073727 \pm 0.016194$ | $0.043749 \pm 0.009696$ |
| | Rel.Red.(%) | 4.70% ↓ | 4.14% ↓ | 4.24% ↓ | 4.98% ↓ | 6.77% ↓ | 5.32% ↓ |

eterization, neither variant uses the physics-guided regularization loss $\mathcal{L}_{phys}$ in this diagnostic. For each held-out sample, both methods are initialized from the same noise $X_0$, and their intermediate rollout states are recorded to compute the normalized off-path deviation $D_\perp(t)$ from the ideal rectified-flow path, where lower values indicate less trajectory drift. As shown in Table 4, x-pred consistently achieves smaller $D_\perp(t)$ than v-pred across all datasets and rollout steps. This provides direct trajectory-level evidence that data-space parameterization suppresses rollout drift and keeps the sampling trajectory closer to the ideal RF path. We further compare the SSIM evolution over forecast lead times in Fig. 6. v-pred does not exhibit a smooth or coherent degradation pattern over the prediction horizon. Instead, its SSIM curves fluctuate irregularly, with abrupt drops and recoveries across lead times, suggesting that structural information is not preserved in a temporally consistent manner during long-term rollout. In contrast, x-pred shows a more gradual and stable decline in SSIM, with substantially reduced temporal oscillations, indicating more reliable structural preservation as the forecast horizon increases. Together, the off-path diagnostic and SSIM curves demonstrate that the advantage of x-pred is not only reflected in reduced geometric deviation during rollout, but also in improved temporal consistency of forecast morphology.

## 5. Conclusion

This paper presents PDRF, a precipitation nowcasting method for addressing the instability of Rectified Flow in long-horizon physical forecasting. The proposed method reformulates the generative ODE with a data-space parameterization and analytically induces the vector field from predicted clean future states, thereby suppressing off-manifold drift during sampling. It further incorporates a Semi-Lagrangian advection prior as a soft kinematic constraint to enhance transport coherence and improve the physical plausibility of long-term predictions. Experimental results show that the proposed model outperforms existing

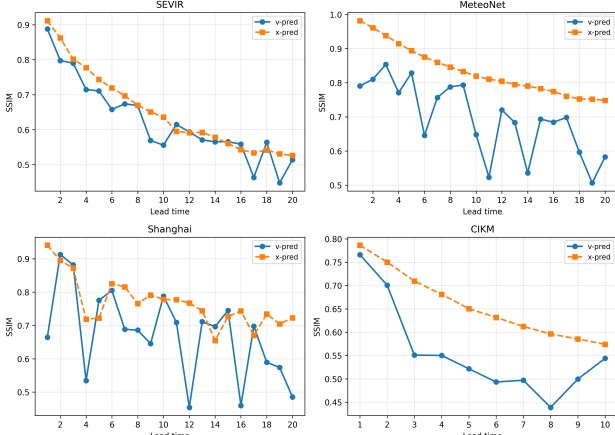

*Figure 6.* SSIM Evolution over Forecast Lead Times. x-pred shows smoother degradation and smaller temporal fluctuations than v-pred, indicating improved long-horizon structural stability.

mainstream methods on public radar benchmarks, validating its effectiveness and reliability for precipitation nowcasting.

## Limitations and Future Work

PDRF achieves competitive performance across different benchmarks, but it still has limitations in modeling complex precipitation dynamics. Since this work focuses on a radar-only setting, auxiliary atmospheric variables such as wind, humidity, temperature, and pressure are not explicitly incorporated, which limits the model's ability to capture processes such as moisture convergence, latent heat release, and rapidly developing convection. Moreover, the Semi-Lagrangian advection prior serves as a lightweight transport constraint rather than a full physical simulator. Future work will explore how to integrate richer atmospheric variables, stronger dynamical priors, and adaptive physical guidance to build a more comprehensive and reliable precipitation nowcasting model.

## Impact Statement

This paper presents work whose goal is to advance the field of Machine Learning. There are many potential societal consequences of our work, none which we feel must be specifically highlighted here.

## Acknowledgements

This work is supported in part by the Yibin Municipal Science and Technology Bureau under Grant 2025JC008.

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

# A. Theoretical Analysis: Stability of Conditional Data-Space Parameterization

In this section, we provide a rigorous derivation demonstrating the superior stability of the Data-Space Parameterization compared to the Velocity-Space Parameterization in the context of *Conditional* Rectified Flow. We explicitly consider the conditioning on historical observations $C$ and analyze the error propagation dynamics of the associated Ordinary Differential Equations (ODEs).

**Preliminaries.** Let $C$ denote the conditional context (past radar frames) and $X_1 \sim \pi(\cdot|C)$ be the target future sequence. The Conditional Rectified Flow defines a linear interpolation path between a noise sample $X_0 \sim \mathcal{N}(0, I)$ and the target $X_1$:

$$Z_t = tX_1 + (1-t)X_0, \quad t \in [0, 1]. \quad (15)$$

The ideal velocity field $v^*(Z_t, t, C)$ governing this flow satisfies the ODE $dZ_t = v^*(Z_t, t, C)dt$, where $v^* = X_1 - X_0$.

We compare two parameterization choices for the neural network $f_\theta(Z_t, t, C)$:

**1. Velocity-Space Parameterization (V-Space).** The network directly predicts the flow velocity $\hat{v} = f_\theta(Z_t, t, C)$. The generative ODE is:

$$\frac{dZ_t}{dt} = f_\theta(Z_t, t, C). \quad (16)$$

**2. Data-Space Parameterization (D-Space).** The network predicts the clean target sequence $\hat{X}_1 = f_\theta(Z_t, t, C)$. The velocity is derived analytically from the linear interpolation geometry $X_1 = Z_t + (1-t)v$. Solving for $v$ yields the derived velocity field:

$$\hat{v}_{data}(Z_t, t, C) = \frac{\hat{X}_1(Z_t, t, C) - Z_t}{1 - t}. \quad (17)$$

Substituting this into the flow equation, the generative ODE becomes:

$$\frac{dZ_t}{dt} = \frac{f_\theta(Z_t, t, C) - Z_t}{1 - t}. \quad (18)$$

**Connection to standard RF and numerical stabilization.** Under the linear coupling $Z_t = (1-t)X_0 + tX_1$, we have

$$X_1 - Z_t = (1-t)(X_1 - X_0), \quad (19)$$

and thus the induced target velocity satisfies

$$\frac{X_1 - Z_t}{1 - t} = X_1 - X_0. \quad (20)$$

Therefore, when the time factor is chosen as $\bar{t} = 1 - t$, the data-induced velocity target used in our formulation is

algebraically identical to the standard rectified-flow target. In practice, we use the stabilized factor $\bar{t} = \max(1-t, \varepsilon)$ to avoid numerical explosion as $t \to 1$, which only modifies the dynamics within an $\varepsilon$-neighborhood of the terminal time.

**Perturbation Analysis and Error Dynamics.** Let $Z_t^*$ be the optimal trajectory generated by the ideal model, and let $Z_t = Z_t^* + \epsilon_t$ be the actual trajectory with a small perturbation error $\epsilon_t$. We analyze the time evolution of this error, $\frac{d\epsilon_t}{dt}$.

**Case 1: Instability of Velocity-Space Parameterization.** Substituting the perturbed state $Z_t^* + \epsilon_t$ into the V-Space ODE and applying a first-order Taylor expansion around $Z_t^*$:

$$\frac{d(Z_t^* + \epsilon_t)}{dt} = f_\theta(Z_t^* + \epsilon_t, t, C) \quad (21)$$

$$\approx f_\theta(Z_t^*, t, C) + \nabla_z f_\theta(Z_t^*, t, C) \cdot \epsilon_t. \quad (22)$$

Subtracting the unperturbed motion $\frac{dZ_t^*}{dt} = f_\theta(Z_t^*, t, C)$, we obtain the error dynamics:

$$\frac{d\epsilon_t}{dt} = \mathbf{J}_v(t, C)\epsilon_t, \quad (23)$$

where $\mathbf{J}_v(t, C) = \nabla_z f_\theta(\cdot)$ is the Jacobian of the predicted velocity field. *Analysis:* In precipitation nowcasting, the velocity field represents highly nonlinear advection and turbulent deformation. Consequently, $\mathbf{J}_v$ often contains large positive eigenvalues (corresponding to divergent or shearing regions). This causes the error norm $\|\epsilon_t\|$ to grow exponentially over time, leading to the "off-manifold drift" observed in standard RF models. There is no explicit term to suppress this growth.

**Case 2: Stability of Data-Space Parameterization (Restoring Force).** Substituting the perturbed state into the D-Space ODE (Eq. 18):

$$\frac{d(Z_t^* + \epsilon_t)}{dt} = \frac{f_\theta(Z_t^* + \epsilon_t, t, C) - (Z_t^* + \epsilon_t)}{1 - t}. \quad (24)$$

We expand the network prediction $f_\theta$ (which predicts $\hat{X}_1$) via Taylor series:

$$f_\theta(Z_t^* + \epsilon_t, t, C) \approx f_\theta(Z_t^*, t, C) + \mathbf{J}_x(t, C)\epsilon_t, \quad (25)$$

where $\mathbf{J}_x(t, C) = \nabla_z f_\theta(\cdot)$ is the Jacobian of the predicted data denoising function. Substituting this back and subtracting the ideal motion $\frac{dZ_t^*}{dt} = \frac{f_\theta(Z_t^*, t, C) - Z_t^*}{1-t}$:

$$\frac{d\epsilon_t}{dt} = \frac{[f_\theta(Z_t^*, t, C) + \mathbf{J}_x\epsilon_t] - (Z_t^* + \epsilon_t)}{1 - t}$$
$$- \frac{f_\theta(Z_t^*, t, C) - Z_t^*}{1 - t} = \frac{\mathbf{J}_x\epsilon_t - \epsilon_t}{1 - t}$$
$$= -\frac{1}{1-t}(I - \mathbf{J}_x)\epsilon_t. \quad (26)$$

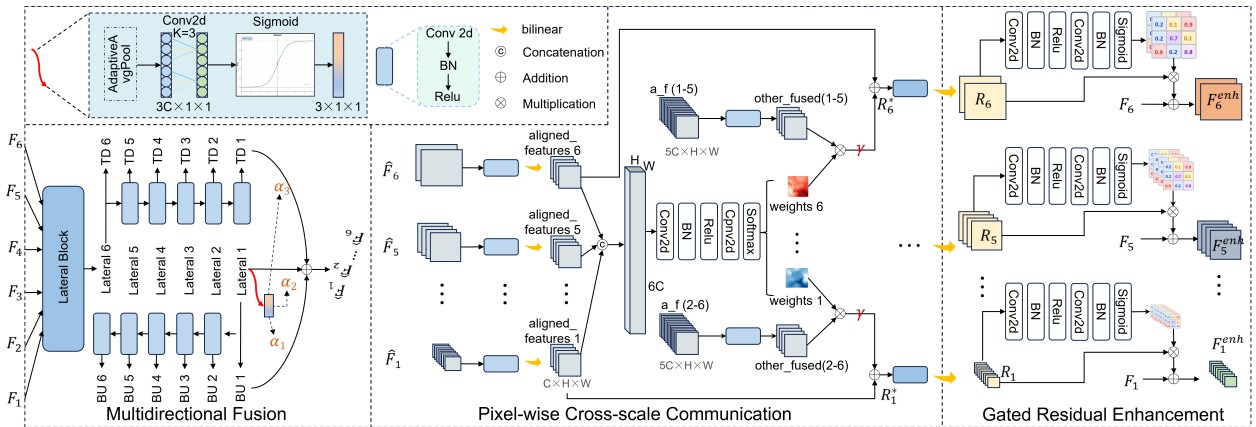

*Figure 7.* **Feature Communication Module (FCM).** *Left:* Multi-directional fusion—top–down (TD), bottom–up (BU), and lateral branches are built at every level; an SE head conditioned on $\mathbf{F}_i^{\text{lat}}$ produces weights $\boldsymbol{\alpha}_i$ that mix the three streams into $\widehat{\mathbf{F}}_i$. *Middle:* Pixel-wise cross-scale communication—all $\widehat{\mathbf{F}}_i$ are first projected with $1 \times 1$ convolutions and bilinearly aligned to a reference resolution $(H^\star \times W^\star)$, concatenated, and fed to the attention head $g$ to obtain per-pixel scale softmax weights $\mathbf{W}_{\text{att}}$. For each level $i$, features from the other scales are fused by a convolutional head and gated by $\mathbf{W}_{\text{att}}[i]$ to yield a per-level enhanced map $\mathbf{R}_i^\star$ at the reference resolution. *Right:* Gated residual enhancement—each $\mathbf{R}_i^\star$ is adapted by the mapping head $\varphi_i$ to $\mathbf{R}_i$, modulated by the sigmoid gate map $\mathbf{G}_i$, and fused residually with the original features $\mathbf{F}_i$ to produce $\mathbf{F}_i^{\text{out}}$.

*Analysis:* Eq. (26) exposes an explicit attraction term scaled by $\frac{1}{1-t}$. To interpret it geometrically, decompose the perturbation into tangent and normal components with respect to the data manifold at $Z_t^*$: $\epsilon_t = \epsilon_t^{\parallel} + \epsilon_t^{\perp}$. For a map that behaves locally like a manifold projector/denoiser, it is natural to assume that $f_\theta$ is approximately identity along tangent directions and contractive along normal directions, i.e., $\mathbf{J}_x \epsilon^{\parallel} \approx \epsilon^{\parallel}$ and $\|\mathbf{J}_x \epsilon^{\perp}\| \le \rho \|\epsilon^{\perp}\|$ for some $\rho < 1$. Under this assumption, Eq. (26) yields $\frac{d\epsilon_t^{\perp}}{dt} \approx -\frac{1-\rho}{1-t} \epsilon_t^{\perp}$, so the off-manifold (normal) error is strongly damped as $t \to 1$ (up to the numerical cutoff $\varepsilon$), while tangent components need not vanish. This explains why the data-space parameterization suppresses off-manifold drift near the terminal time.

## B. Supplementary Algorithms and Architectural Details

### B.1. Algorithms

In this section, we provide the detailed pseudocode for the training and inference procedures of the proposed PDRF framework. Algorithm 1 outlines the training step, which integrates the data-space flow matching objective with the physics-guided regularization derived from Semi-Lagrangian extrapolation. Algorithm 2 details the conditional inference process using a stabilized Euler method to generate future radar sequences from the learned vector field.

---

**Algorithm 1** Training step (Conditional Data-space Rectified Flow with Physics Regularization)

---

**Require:** Condition $C$, target $X_1$, network $f_\theta(\cdot)$, weight $\lambda_{\text{phy}}$, stabilizer $\varepsilon$
**Ensure:** Total loss $\mathcal{L}_{\text{total}}$
1: Sample flow time $t \sim \mathcal{U}[0, 1]$ and noise $X_0 \sim \mathcal{N}(0, I)$
2: Compute stabilized time factor $\bar{t} \leftarrow \max(1 - t, \varepsilon)$
3: Construct intermediate state $Z_t \leftarrow t X_1 + (1 - t) X_0$
4: Predict clean data $\hat{X}_1 \leftarrow f_\theta(Z_t, t, C)$
5: Derive implied velocity $v_{\text{pred}} \leftarrow (\hat{X}_1 - Z_t)/\bar{t}$
6: Compute target velocity $v_{\text{tgt}} \leftarrow (X_1 - Z_t)/\bar{t}$
7: Generate physics prior $X_{\text{SL}} \leftarrow \text{SemiLagPrior}(C)$
8: Map prior to velocity space $v_{\text{SL}} \leftarrow (X_{\text{SL}} - Z_t)/\bar{t}$
9: Compute flow matching loss $\mathcal{L}_{\text{flow}} \leftarrow \|v_{\text{pred}} - v_{\text{tgt}}\|_2^2$
10: Compute physics regularization $\mathcal{L}_{\text{phy}} \leftarrow \|v_{\text{pred}} - v_{\text{SL}}\|_2^2$
11: $\mathcal{L}_{\text{total}} \leftarrow \mathcal{L}_{\text{flow}} + \lambda_{\text{phy}} \mathcal{L}_{\text{phy}}$
12: **return** $\mathcal{L}_{\text{total}}$

---

### B.2. Network Architecture Details

The generator adopts a U-KAN backbone, which is based on a four-scale U-Net architecture. A conditional encoder processes $\mathbf{X}_{t-J+1:t}$ and produces multi-scale features $\{\mathbf{F}_i\}_{i=1}^L$ (from shallow to deep). The framework integrates specialized modules to address nowcasting challenges: FCM performs bidirectional cross–scale communication before decoding; CGSTF operates at shallow stages (stages 1–3) to reduce inter-frame misalignment by aligning backbone features; WGSC acts on deep and bottleneck skip connections to modulate fusion via wavelet-derived gates; and

**Algorithm 2** Inference Sampling (Stabilized Euler Method)

---

**Require:** Condition $C$, network $f_\theta(\cdot)$, steps $N$, stabilizer $\varepsilon$
**Ensure:** Predicted sequence $\hat{X}_1$

1: Define time steps $0 = t_0 < t_1 < \cdots < t_N = 1 - \varepsilon$
2: Initialize $Z \sim \mathcal{N}(0, I)$
3: **for** $i = 0$ to $N - 1$ **do**
4:      $\bar{t} \leftarrow \max(1 - t_i, \varepsilon)$
5:      Predict clean data $\hat{X}_1 \leftarrow f_\theta(Z, t_i, C)$
6:      Derive velocity $v_{\text{pred}} \leftarrow (\hat{X}_1 - Z)/\bar{t}$
7:      Update state $Z \leftarrow Z + (t_{i+1} - t_i)\, v_{\text{pred}}$
8: **end for**
9: Final refinement $\hat{X}_1 \leftarrow f_\theta(Z, 1 - \varepsilon, C)$
10: **return** $\hat{X}_1$

---

KAN-VRWKV blocks are placed at the deepest stages to supply long-range spatiotemporal context, where VRWKV serves as the temporal-mixing unit. See Fig. 7 for the full schematic.

### B.2.1. FEATURE COMMUNICATION MODULE (FCM)

Radar echo sequences contain broad motion patterns and rapidly evolving local echoes. Shallow features emphasize spatial localization and detail, whereas deep features summarize global context and longer-range dependencies. Simple concatenation or summation across scales can leave these cues weakly coupled, making it difficult to propagate global context to fine-grained localization. To address this, we construct a full-scale communication hub: multi-directional pathways connect pyramid levels in top-down, bottom-up, and lateral directions, and pixel-wise cross-scale selection enables each location to draw complementary information from the most relevant scale. An overview of FCM is shown in Fig. 7.

**Multidirectional fusion.** Let $\mathbf{F}_i$ denote the feature map at scale $i$, where $i \in \{1, \ldots, L\}$ indexes pyramid levels from the shallowest ($i = 1$) to the deepest ($i = L$). FCM constructs three distinct pathways:

$$\mathbf{F}_i^{\text{td}} = \mathcal{U}\big(\phi_i^{\text{td}}(\mathbf{F}_{i+1}^{\text{td}})\big), \quad \mathbf{F}_L^{\text{td}} = \phi_L^{\text{lat}}(\mathbf{F}_L), \quad (27)$$

$$\mathbf{F}_i^{\text{bu}} = \mathcal{D}\big(\phi_{i-1}^{\text{bu}}(\mathbf{F}_{i-1}^{\text{bu}})\big), \quad \mathbf{F}_1^{\text{bu}} = \phi_1^{\text{lat}}(\mathbf{F}_1), \quad (28)$$

$$\mathbf{F}_i^{\text{lat}} = \phi_i^{\text{lat}}(\mathbf{F}_i), \quad (29)$$

where $\phi^{(\cdot)}$ are Conv–BN–ReLU blocks with $1\times1$ or $3\times3$ kernels, and $\mathcal{U}$ and $\mathcal{D}$ denote bilinear upsampling and downsampling, respectively. The three paths are combined per scale via learned weights $\boldsymbol{\alpha}_i \in \mathbb{R}^3$ derived from a squeeze-and-excitation (SE) operation on $\mathbf{F}_i^{\text{lat}}$:

$$\widehat{\mathbf{F}}_i = \sum_{j \in \{\text{td,bu,lat}\}} \alpha_{i,j}\, \mathbf{F}_i^j, \quad \boldsymbol{\alpha}_i = \text{softmax}\big(\psi(\mathbf{F}_i^{\text{lat}})\big). \quad (30)$$

**Pixel-wise cross-scale communication.** All multi-directional features $\widehat{\mathbf{F}}_i$ are first aligned to a common reference resolution (the middle scale) via learned projections, producing a set of aligned features $\{\widetilde{\mathbf{F}}_i\}_{i=1}^L$. These are concatenated and processed by an attention head $g$ to compute per-pixel scale weights:

$$\mathbf{W}_{\text{att}} = \text{softmax}_{\text{scale}}\Big(g\big(\text{Concat}(\widetilde{\mathbf{F}}_1, \ldots, \widetilde{\mathbf{F}}_L)\big)\Big) \in \mathbb{R}^{L \times H^\star \times W^\star}. \quad (31)$$

To facilitate direct, per-scale communication, for each reference feature map $\widetilde{\mathbf{F}}_i$, we aggregate all *other* maps via a fusion convolution $\mathbf{F}_{\text{other},i} = \phi_{\text{other}}\big(\text{Concat}(\{\widetilde{\mathbf{F}}_j\}_{j \neq i})\big)$ and compute the enhanced feature residually:

$$\mathbf{R}_i^\star = \widetilde{\mathbf{F}}_i + \gamma\,\big(\mathbf{W}_{\text{att}}[i] \odot \mathbf{F}_{\text{other},i}\big), \quad (32)$$

where $\gamma$ is a learnable scalar.

**Gated residual enhancement.** The enhanced features $\{\mathbf{R}_i^\star\}$ are routed back to each native level $i$. A projection head $\varphi_i$ adapts each $\mathbf{R}_i^\star$ to its native dimensionality:

$$\mathbf{R}_i = \varphi_i\big(\text{Resize}(\mathbf{R}_i^\star \to i)\big) \in \mathbb{R}^{C_i \times H_i \times W_i}. \quad (33)$$

To avoid indiscriminate injection of cross-scale context, a pixel- and channel-wise gating branch produces a dense mask $\mathbf{G}_i$:

$$\mathbf{G}_i = \sigma\Big(\text{BN}\big(\text{Conv}_{1\times1}(\text{BN}(\text{ReLU}(\text{Conv}_{3\times3}(\mathbf{R}_i))))\big)\Big)$$
$$\in [0,1]^{C_i \times H_i \times W_i}. \quad (34)$$

The final output is obtained by gated residual fusion:

$$\mathbf{F}_i^{\text{out}} = \mathbf{F}_i + \mathbf{G}_i \odot \mathbf{R}_i, \quad (35)$$

where $\odot$ denotes element-wise multiplication.

### B.2.2. CONDITION-GUIDED SPATIAL TRANSFORM FUSION (CGSTF)

The evolution of precipitation systems often involves complex, non-rigid motion, leading to significant spatial displacement between consecutive radar frames. This causes intermediate features in the main backbone to be spatially misaligned with those from the conditional encoder. To mitigate this, CGSTF learns to predict a displacement field directly from the conditioning echoes and applies a differentiable geometric transform to align the backbone features prior to fusion. An overview of CGSTF is shown in Fig. 8.

The alignment process begins by generating a displacement field $\mathbf{O}$ from the conditional features $\mathbf{F}_{\text{cond}}$ using a lightweight convolutional network $\phi$. The offsets are bounded for stability using a $\tanh$ function and a scalar hyperparameter $\alpha$:

$$\mathbf{O} = \alpha\,\tanh(\phi(\mathbf{F}_{\text{cond}})) \in \mathbb{R}^{B \times 2 \times H \times W}. \quad (36)$$

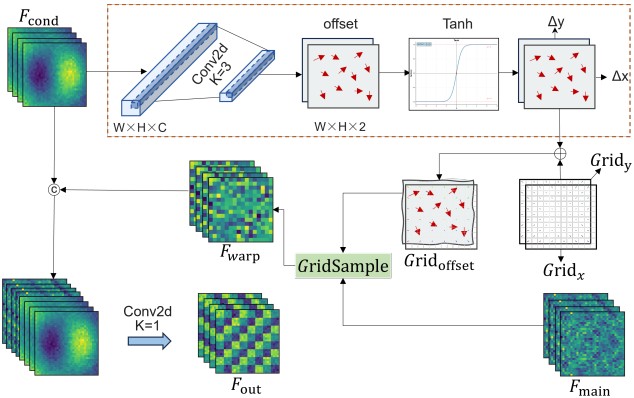

*Figure 8.* **Condition-Guided Spatial Transform Fusion (CGSTF).** Conditioning features produce an offset field $\mathbf{O}$ (bounded by $\tanh$) to perturb the base grid $(\mathrm{Grid}_x, \mathrm{Grid}_y)$ and obtain $\mathrm{Grid}_{\mathrm{offset}}$. GridSample warps the main features to $\mathbf{F}_{\mathrm{warp}}$, which is fused with the conditional features to yield $\mathbf{F}_{\mathrm{out}}$.

This field is added to a normalized base grid $\mathbf{G}_{\mathrm{base}}$ to create the sampling grid $\mathbf{G}_{\mathrm{offset}} = \mathbf{G}_{\mathrm{base}} + \mathbf{O}_{\mathrm{grid}}$. The main backbone features $\mathbf{F}_{\mathrm{main}}$ are then warped via differentiable bilinear sampling:

$$\mathbf{F}_{\mathrm{warp}} = \mathrm{GridSample}\big(\mathbf{F}_{\mathrm{main}}, \mathbf{G}_{\mathrm{offset}}\big). \qquad (37)$$

The warped features are spatially aligned with the conditional features, concatenated, and fused:

$$\mathbf{F}_{\mathrm{out}} = \mathrm{Conv}_{1\times1}\big(\mathrm{Concat}(\mathbf{F}_{\mathrm{warp}}, \mathbf{F}_{\mathrm{cond}})\big). \qquad (38)$$

We deploy CGSTF at stages 1–3 to establish geometric consistency early in the network.

### B.2.3. WAVELET-GUIDED SKIP CONNECTION (WGSC)

Standard skip connections typically pass information indiscriminately. To make the fusion process context-aware, WGSC leverages conditional features as external prior knowledge. Specifically, it employs a 2D discrete wavelet transform (DWT) to decompose features into frequency sub-bands, providing a control mechanism to balance structural information with fine-grained details during fusion. An overview is shown in Fig. 9.

**Wavelet guidance.** Given conditional features $\mathbf{F}_{\mathrm{cond}}$, we first compute the 2D DWT:

$$(\mathbf{F}_{LL}, \mathbf{F}_{LH}, \mathbf{F}_{HL}, \mathbf{F}_{HH}) = \mathrm{DWT}_{\mathrm{2D}}(\mathbf{F}_{\mathrm{cond}}). \qquad (39)$$

The structural ($LL$) and detail ($LH, HL, HH$) components are processed by separate stems, upsampled, and fused to produce a unified wavelet guidance map $\mathbf{A}_{\mathrm{wav}}$:

$$\mathbf{A}_{\mathrm{wav}} = \sigma\Big(\phi_{\mathrm{fuse}}\big(\mathrm{Concat}(\mathbf{Q}_{\mathrm{low}}^{\uparrow}, \mathbf{Q}_{\mathrm{high}}^{\uparrow})\big)\Big) \in [0,1]^{C \times H \times W}. \qquad (40)$$

**Adaptive skip fusion.** We synthesize three gates—spatial ($\mathbf{M}_s$), channel ($\mathbf{M}_c$), and wavelet ($\mathbf{M}_w$)—from the concatenation of encoder features, decoder features, and $\mathbf{A}_{\mathrm{wav}}$. These gates modulate the respective streams:

$$\mathbf{F}'_{\mathrm{enc}} = \mathbf{F}_{\mathrm{enc}} \odot \mathrm{Broadcast}(\mathbf{M}_s) \odot \mathrm{Broadcast}(\mathbf{M}_c) \qquad (41)$$

$$\mathbf{F}'_{\mathrm{dec}} = \mathbf{F}_{\mathrm{dec}} \odot \mathbf{M}_w \qquad (42)$$

The modulated features are combined via a learned dynamic fusion weight $\omega$:

$$\mathbf{F}_{\mathrm{fused}} = \omega \odot \phi_{\mathrm{enc\_proc}}(\mathbf{F}'_{\mathrm{enc}}) + (1-\omega) \odot \phi_{\mathrm{dec\_proc}}(\mathbf{F}'_{\mathrm{dec}}), \qquad (43)$$

where $\omega = \sigma(\phi_{\mathrm{fusion}}(\mathrm{Concat}(\dots)))$. Finally, the output is obtained by refining the concatenation of the fused and original features:

$$\mathbf{F}_{\mathrm{out}} = \phi_{\mathrm{out}}(\mathrm{Concat}(\mathbf{F}_{\mathrm{fused}}, \mathbf{F}_{\mathrm{enc}})). \qquad (44)$$

This allows the network to combine structural details from the encoder with context-aware decoder features, guided by frequency-domain priors.

### B.2.4. VRWKV FOR EFFICIENT LONG-RANGE MODELING

As forecast horizons and spatial resolutions increase, standard self-attention becomes a computational bottleneck. To address this, we place lightweight VRWKV blocks at deep stages—encoder tail, bottleneck, and first decoder layer—where spatial resolution is lower and receptive fields are larger. VRWKV mixes tokens using linear-time sequence kernels while remaining parallelizable, extending temporal context modeling capabilities at near-linear cost. We adopt a serial integration strategy for these blocks (as illustrated in Fig. 10b), which our ablation studies confirmed as the optimal configuration.

## C. Detailed Ablation Studies on Module Components

In this section, we provide a granular analysis of the internal components of the proposed modules (FCM, CGSTF, WGSC) and the deployment strategies for the spatiotemporal memory unit (VRWKV). These experiments validate the design choices made for the specific challenges of precipitation nowcasting.

### C.1. Dissection of the Feature Communication Module (FCM)

The FCM consists of three main components: multidirectional fusion, cross-scale attention, and gated residual enhancement. As shown in Table 5, we performed an ablation study on the MeteoNet dataset. Removing any component degrades performance, confirming their complementary roles.

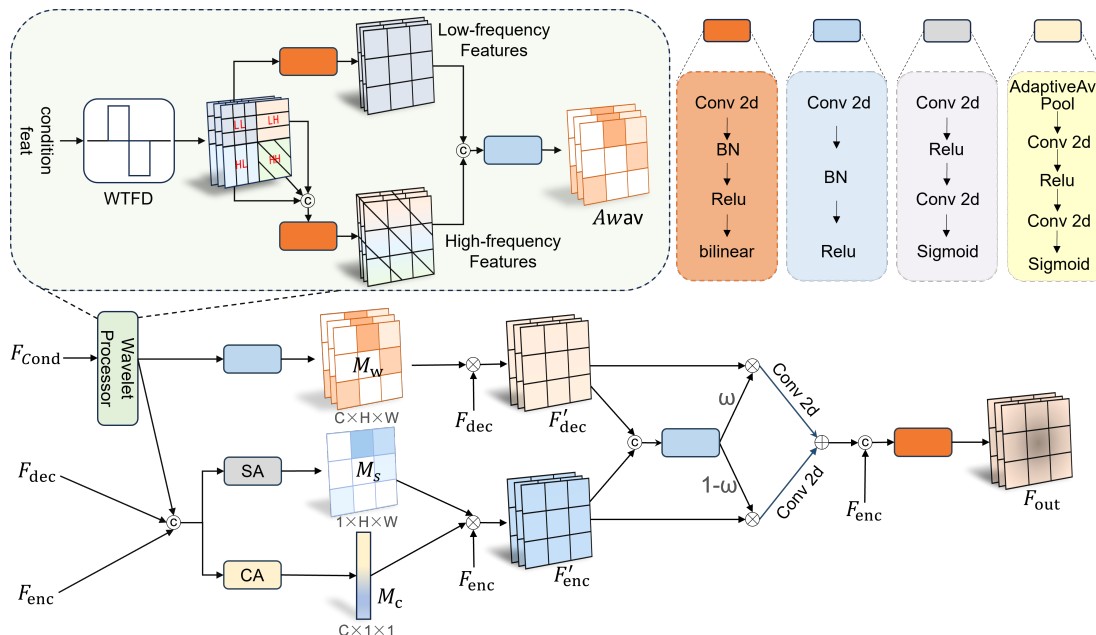

*Figure 9.* **The Wavelet-Guided Skip Connection (WGSC) architecture.** Conditional features are processed by the Wavelet Processor (top inset) via 2D DWT to yield a frequency-aware guidance map ($\mathbf{A}_{\text{wav}}$). Concurrently (main path, bottom), spatial (SA) and channel (CA) attention masks are derived from concatenated encoder and decoder features. These components, along with $\mathbf{A}_{\text{wav}}$, inform the synthesis of three distinct gates. Encoder features are modulated by SA/CA gates, while decoder features are modulated by a wavelet-derived gate. An adaptive fusion mechanism combines these modulated streams based on a learned weight $\omega$, followed by a final convolutional refinement step to produce the output.

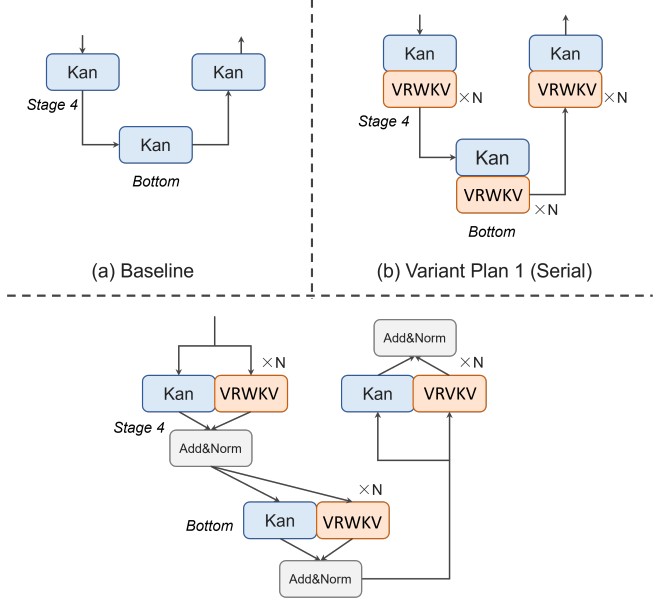

*Figure 10.* **VRWKV placement strategies at deep stages.** Comparison of integration strategies for VRWKV blocks at the encoder tail, bottleneck, and first decoder layer. **(a)** Backbone architecture at these locations, without VRWKV. **(b)** Serial insertion with $N$ blocks per stage; specifically, the $N = 1$ case corresponds to the KAN-VRWKV Block configuration used in PDRF. **(c)** Parallel integration via residual mixing (Add&Norm).

*Table 5.* Ablation study on FCM components on MeteoNet dataset. We analyze the contribution of each component within the FCM module. The best results are shown in **bold**.

| Method | CSI-M↑ | CSI-24↑ | CSI-32↑ | HSS↑ | MSE↓ |
|---|---|---|---|---|---|
| w/o Cross Attention | 0.4089 | 0.3945 | 0.2334 | 0.5412 | 12.43 |
| w/o Multi Fusion | 0.4034 | 0.3889 | 0.2298 | 0.5367 | 12.56 |
| w/o CA & MF | 0.3987 | 0.3823 | 0.2245 | 0.5298 | 12.78 |
| w/o Enhancement | 0.4178 | 0.4045 | 0.2423 | 0.5521 | 12.08 |
| **Full FCM** | **0.4213** | **0.4087** | **0.2477** | **0.5563** | **11.91** |

Disabling multi-directional fusion causes the most significant drop in high-threshold CSI (a decrease of about 8% for CSI-32), highlighting the critical importance of bidirectional information flow for coordinating global context and local details. Removing the cross-scale attention mechanism also leads to a notable performance decline, particularly for high-intensity events, which indicates that the per-pixel adaptive scale selection is crucial for retaining sharp, well-defined structures. The final enhancement stage provides a smaller but consistent benefit, helping to effectively integrate the globally-aware features back into each scale. The full FCM configuration achieves the best overall performance, supporting our hypothesis that a holistic communication strategy is superior to simpler fusion methods.

## C.2. Analysis of the Condition-Guided Spatial Transform Fusion (CGSTF)

We evaluated alternative fusion strategies to CGSTF on the SEVIR dataset, with results presented in Table 6. Replacing our explicit alignment-and-fusion approach with simple element-wise addition or concatenation leads to a significant drop in performance, especially at the highest threshold (CSI-219 decreases by 8.5% and 6.2%, respectively). This confirms that directly merging spatially misaligned features is suboptimal.

*Table 6.* Ablation study on CGSTF components on SEVIR dataset. We analyze different fusion strategies and key components within the module. The best results are shown in **bold**.

| Method | CSI-M↑ | CSI-181↑ | CSI-219↑ | HSS↑ | MSE↓ |
|---|---|---|---|---|---|
| w/ Addition | 0.3478 | 0.1889 | 0.0987 | 0.4456 | 441.23 |
| w/ Concatenation | 0.3501 | 0.1923 | 0.1012 | 0.4489 | 439.78 |
| w/o Offset Network | 0.3467 | 0.1876 | 0.0967 | 0.4423 | 443.91 |
| w/o Tanh Activation | 0.3523 | 0.1934 | 0.1045 | 0.4512 | 438.67 |
| **Full CGSTF** | **0.3552** | **0.1966** | **0.1079** | **0.4576** | **435.54** |

The most substantial degradation occurs when the offset prediction network is removed entirely (CSI-219 drops by 10.4%), underscoring that the learned, data-driven displacement field is the key mechanism for improving high-threshold skill. Furthermore, removing the tanh activation, which bounds the predicted offsets, also results in a consistent, albeit smaller, performance drop, suggesting that this constraint is important for maintaining stable and physically plausible warping. These findings collectively validate that the explicit, bounded, and condition-guided warping performed by CGSTF is a more effective strategy than direct feature merging.

## C.3. Evaluation of Wavelet-Guided Skip Connection (WGSC)

Table 7 details the ablation of WGSC components on the CIKM dataset. Removing the wavelet processor—the core of the module—results in a significant performance drop, particularly for high-intensity rainfall (CSI-40 decreases by 7.4%). This strongly suggests that the frequency-based decomposition into structural (low-frequency) and detail (high-frequency) cues is effective for guiding the reconstruction process.

*Table 7.* Ablation study on WGSC module components on CIKM dataset. We analyze the contribution of key components. The best results are shown in **bold**.

| Method | CSI-M↑ | CSI-35↑ | CSI-40↑ | HSS↑ | MSE↓ |
|---|---|---|---|---|---|
| w/o Wavelet Processor | 0.3198 | 0.2298 | 0.1543 | 0.4156 | 44.67 |
| w/o Condition Attention | 0.3234 | 0.2356 | 0.1598 | 0.4234 | 44.23 |
| w/o Adaptive Fusion | 0.3167 | 0.2267 | 0.1523 | 0.4089 | 44.89 |
| **Full WGSC** | **0.3293** | **0.2406** | **0.1666** | **0.4278** | **43.16** |

Disabling the condition attention mechanism also leads to a consistent decline in skill, indicating that the gates must be adaptive to the specific conditional input to be fully effective. The largest overall degradation occurs when the final adaptive fusion stage is removed, highlighting the importance of the mechanism that integrates the wavelet cues with the encoder and decoder features. The results confirm that all three components of WGSC—wavelet decomposition, conditional attention, and adaptive fusion—contribute in a complementary manner to its success.

## C.4. Optimal VRWKV Deployment Strategy

We explored both serial and parallel integration strategies for the VRWKV module at different depths, with results on the Shanghai dataset shown in Table 8.

*Table 8.* Ablation study on VRWKV deployment strategies on Shanghai dataset. The best results are shown in **bold**.

| Method | CSI-M↑ | CSI-35↑ | CSI-40↑ | HSS↑ | MSE↓ |
|---|---|---|---|---|---|
| *Parallel Integration* | | | | | |
| Parallel 1-layer | 0.4121 | 0.3765 | 0.2736 | 0.5475 | 32.31 |
| Parallel 2-layer | 0.4085 | 0.3716 | 0.2651 | 0.5422 | 31.49 |
| Parallel 3-layer | 0.4115 | 0.3763 | 0.2731 | 0.5483 | 33.01 |
| *Serial Integration* | | | | | |
| **Serial 1-layer** | **0.4198** | **0.3867** | **0.2823** | **0.5583** | **32.91** |
| Serial 2-layer | 0.4088 | 0.3713 | 0.2639 | 0.5434 | 31.49 |
| Serial 3-layer | 0.4114 | 0.3728 | 0.2663 | 0.5467 | 32.58 |
| *Alternative Modules (replacing Serial 1-layer)* | | | | | |
| Linear attention | 0.4158 | 0.3821 | 0.2790 | 0.5521 | 33.05 |
| SE attention | 0.4137 | 0.3794 | 0.2762 | 0.5510 | 33.12 |

A serial three-stage deployment—placing one block at each of the encoder tail, the bottleneck, and the first decoder layer—achieves the best overall skill across metrics. This configuration corresponds to the entry denoted *Serial 1-layer* in the table, where the layer count signifies a single block being applied at each of the three stages. Stacking additional blocks at these same deep stages (i.e., the entries *Serial 2-layer* and *Serial 3-layer*) yields diminishing returns, suggesting that once the most compressed representations have been augmented with long-range context, this single-block-per-stage augmentation is sufficient. In contrast, parallel placement is less consistent; for example, two-branch designs tend to lower MSE while degrading high-threshold CSI, indicating smoother forecasts with attenuated intense echoes. To disentangle architectural effects, we also replaced VRWKV with linear attention and with a channel-only SE block at the bottleneck. Both variants produce similar MSE but systematically underperform on CSI/HSS at medium and high thresholds, implying weaker preservation of fine-grained structure. These observations support our default choice: unless otherwise noted, we adopt the serial one-block, three-stage VRWKV configuration.

*Table 9.* Shanghai ablation on inference sampling steps $N$.

| $N$ | CSI-M | CSI-35 | CSI-40 | HSS | SSIM | MSE |
|---|---|---|---|---|---|---|
| 1 | 0.4314 | 0.3941 | 0.2734 | 0.5644 | 0.7856 | **24.56** |
| 3 | 0.4372 | 0.4014 | 0.3012 | 0.5734 | 0.7958 | 28.38 |
| 5 | **0.4409** | **0.4066** | **0.3032** | **0.5772** | **0.7983** | 30.02 |
| 10 | 0.4360 | 0.4013 | 0.2999 | 0.5726 | 0.7904 | 31.41 |
| 20 | 0.4330 | 0.3977 | 0.2974 | 0.5696 | 0.7865 | 31.99 |
| 50 | 0.4301 | 0.3941 | 0.2939 | 0.5666 | 0.7817 | 32.09 |
| 100 | 0.4292 | 0.3931 | 0.2932 | 0.5658 | 0.7796 | 32.12 |

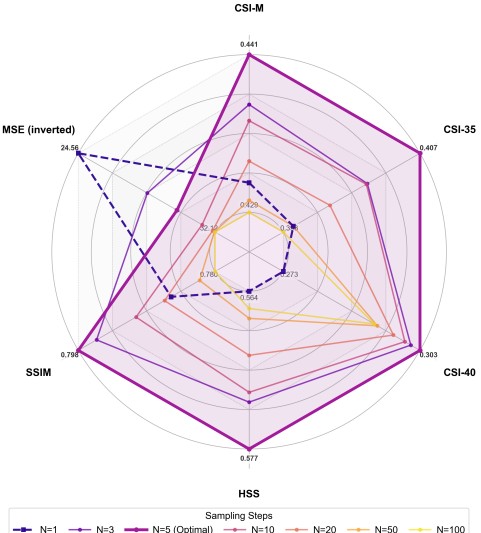

*Figure 11.* Radar plot showing the performance trade-off across different sampling steps $N$ ($N \in \{1, 3, 5, 10, 20, 50, 100\}$) in terms of CSI-M, CSI-35, CSI-40, HSS, SSIM, and MSE. For visualization, each metric is min–max normalized over all $N$ and linearly rescaled to the radial range $[0.2, 1.0]$ (leaving a hollow center). MSE is inverted after normalization so that larger radial values consistently indicate better performance. The inner and outer numeric annotations on each axis indicate the worst and best raw values of the corresponding metric, respectively; the curve at $N = 5$ is highlighted as the best-performing setting.

## D. Ablation on Sampling Steps for Deployment (Shanghai)

We ablate the number of inference sampling steps $N$ on the Shanghai test set to identify a practical deployment configuration. Unless otherwise specified, we keep the trained model, data split, and evaluation protocol fixed, and vary only $N \in \{1, 3, 5, 10, 20, 50, 100\}$ at inference time. Table 9 and Fig. 11 report the results.

Overall, $N = 5$ achieves the best performance across the skill-oriented metrics, including CSI-M/CSI-35/CSI-40, HSS, and SSIM. Increasing $N$ beyond 5 does not improve these scores and instead leads to mild degradation, indicating a saturation regime where additional sampling offers diminishing returns. We therefore adopt $N = 5$ as the default inference setting for all Shanghai experiments and

deployment, as it provides the strongest overall forecast quality with minimal additional computational cost.

*Table 10.* Ablation on PercpCast parameterization: velocity-space prediction (v-pred) vs data-space prediction (x-pred).

| Dataset | Method | CSI-M↑ | CSI-$t_1$↑ | CSI-$t_2$↑ | HSS↑ | SSIM↑ | MSE↓ |
|---|---|---|---|---|---|---|---|
| SEVIR | PercpCast-v | 0.3492 | 0.1928 | **0.1022** | 0.4538 | **0.6852** | 449.65 |
| | PercpCast-x | **0.3595** | **0.2016** | 0.1021 | **0.4654** | 0.6817 | **404.73** |
| MeteoNet | PercpCast-v | 0.4038 | 0.3932 | 0.2206 | 0.5436 | 0.8266 | 12.96 |
| | PercpCast-x | **0.4277** | **0.4202** | **0.2433** | **0.5688** | **0.8306** | **12.83** |
| Shanghai | PercpCast-v | 0.3995 | 0.3628 | 0.2577 | 0.5403 | 0.7941 | 39.91 |
| | PercpCast-x | **0.4264** | **0.3939** | **0.2887** | **0.5664** | **0.7957** | **34.58** |
| CIKM | PercpCast-v | 0.3162 | 0.2223 | 0.1553 | 0.4107 | **0.6431** | 45.50 |
| | PercpCast-x | **0.3281** | **0.2312** | **0.1604** | **0.4244** | 0.6415 | **41.44** |

Note: $(t_1, t_2)$ are dataset-specific thresholds: SEVIR (181, 219), MeteoNet (24, 32), Shanghai (35, 40), CIKM (35, 40).

## E. Ablation on Parameterization: Velocity-Space vs. Data-Space Prediction in PercpCast

To study the impact of parameterization in Rectified Flow modeling, we conduct an ablation on PercpCast by switching from velocity-space prediction (v-space, predicting $v$) to data-space prediction (x-space, predicting $x$), i.e., directly forecasting future radar frames in the data space. All other training and sampling settings are kept identical for a fair comparison. As reported in Table 10, x-space prediction consistently improves precipitation event detection performance on all four datasets, leading to higher CSI-M and HSS. Meanwhile, it reduces MSE across the board, indicating improved overall reconstruction accuracy. SSIM slightly increases on MeteoNet and Shanghai, while remaining comparable with minor fluctuations on SEVIR and CIKM. We additionally provide qualitative results on SEVIR in Fig. 12. The two parameterizations produce broadly similar large-scale precipitation patterns, and both capture the main evolution trend over the forecasting horizon. In these examples, the x-space variant tends to better maintain the spatial continuity of precipitation structures and retain weak echoes at longer lead times, which is consistent with its improved quantitative scores. Overall, the visual comparison corroborates the table results and supports the effectiveness of data-space prediction for radar nowcasting.

## F. Additional Analysis of AMSE and Temporal Spectral Characteristics

We further investigate AMSE-based spectral loss adjustment from both quantitative and frequency-domain perspectives. As shown in Table 11, applying AMSE (Subich et al., 2025) to deterministic models improves the plain Backbone on several metrics, suggesting that emphasizing predictable spectral scales is beneficial for deterministic nowcasting. However, the gains are architecture- and dataset-dependent; for example, AlphaPre+AMSE degrades on Shanghai and

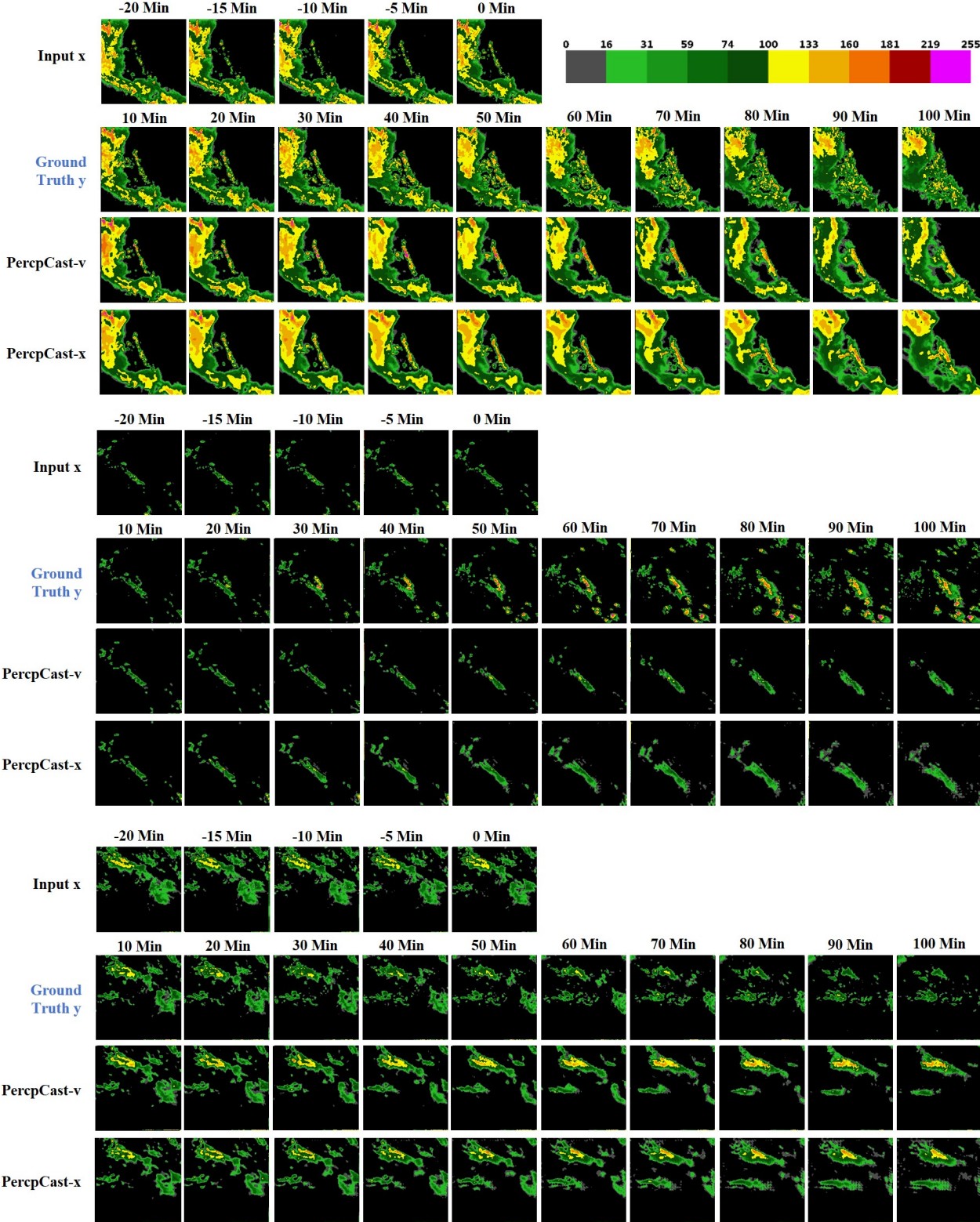

*Figure 12.* Qualitative comparisons on the SEVIR dataset: PercpCast-v vs. PercpCast-x.

*Table 11.* Comparison with deterministic models trained with AMSE loss on four radar nowcasting datasets. The best results are highlighted in bold and the second-best results are underlined.

| Method | SEVIR | | | | | | MeteoNet | | | | | |
|---|---|---|---|---|---|---|---|---|---|---|---|---|
| | CSI-M↑ | CSI-181↑ | CSI-219↑ | HSS↑ | SSIM↑ | MSE↓ | CSI-M↑ | CSI-24↑ | CSI-32↑ | HSS↑ | SSIM↑ | MSE↓ |
| Backbone | 0.3095 | 0.1049 | 0.0401 | 0.3875 | 0.6329 | 461.26 | 0.3793 | 0.3453 | 0.1601 | 0.5026 | 0.8059 | 13.31 |
| Backbone + AMSE | 0.3243 | 0.1526 | 0.0759 | 0.4161 | 0.6121 | 401.16 | 0.4079 | 0.3904 | 0.2102 | 0.5438 | 0.8232 | **10.27** |
| AlphaPre | 0.3259 | 0.1332 | 0.0545 | 0.4110 | **0.6884** | **345.18** | 0.3824 | 0.3633 | 0.2002 | 0.5164 | 0.7968 | 12.74 |
| AlphaPre + AMSE | 0.3349 | 0.1572 | 0.0665 | 0.4225 | 0.6230 | 396.33 | 0.3841 | 0.3655 | 0.1926 | 0.5123 | 0.7956 | 11.43 |
| PDRF (Ours) | **0.3622** | **0.2110** | **0.1199** | **0.4678** | 0.6869 | 451.80 | **0.4386** | **0.4288** | **0.2684** | **0.5747** | **0.8385** | 11.14 |

| Method | Shanghai | | | | | | CIKM | | | | | |
|---|---|---|---|---|---|---|---|---|---|---|---|---|
| | CSI-M↑ | CSI-35↑ | CSI-40↑ | HSS↑ | SSIM↑ | MSE↓ | CSI-M↑ | CSI-35↑ | CSI-40↑ | HSS↑ | SSIM↑ | MSE↓ |
| Backbone | 0.4002 | 0.3550 | 0.2420 | 0.5260 | 0.6583 | 34.01 | 0.2988 | 0.2023 | 0.1276 | 0.3849 | 0.6043 | 46.99 |
| Backbone + AMSE | 0.4068 | 0.3671 | 0.2538 | 0.5384 | 0.7755 | 28.89 | 0.3199 | 0.2203 | 0.1479 | 0.4109 | 0.6155 | 40.28 |
| AlphaPre | 0.4178 | 0.3854 | 0.2615 | 0.5534 | 0.7951 | **28.02** | 0.3194 | 0.2068 | 0.1416 | 0.4137 | 0.6568 | **35.18** |
| AlphaPre + AMSE | 0.3478 | 0.3071 | 0.1874 | 0.4743 | 0.6840 | 33.49 | 0.2924 | 0.1907 | 0.1116 | 0.3755 | 0.5655 | 38.88 |
| PDRF (Ours) | **0.4409** | **0.4066** | **0.3032** | **0.5772** | **0.7983** | 30.02 | **0.3329** | **0.2380** | **0.1720** | **0.4312** | **0.6619** | 37.69 |

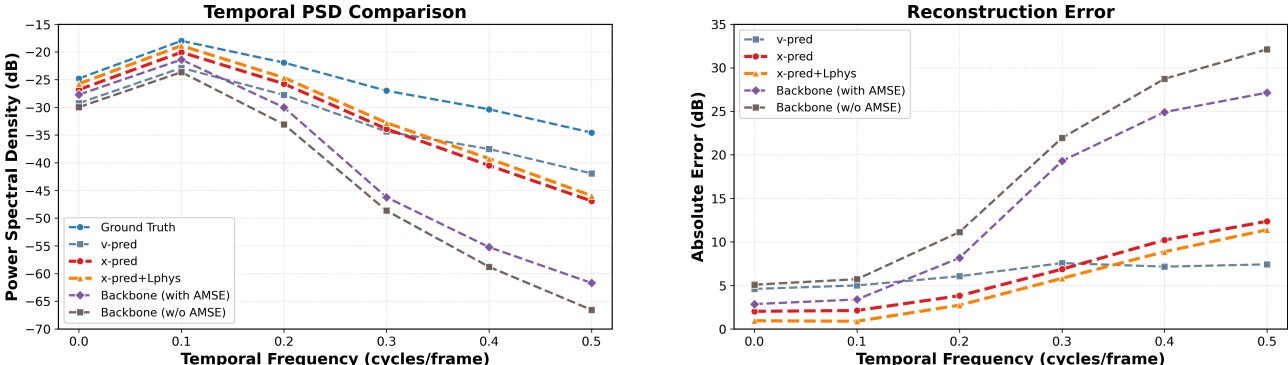

*Figure 13.* Temporal PSD and reconstruction error comparison on SEVIR averaged over 50 samples.

CIKM. In contrast, PDRF consistently achieves the best CSI-M, high-threshold CSI, and HSS across all four datasets, demonstrating stronger overall event-based forecasting ability than AMSE-enhanced deterministic models.

To further analyze temporal spectral behavior, we compute the temporal power spectral density (PSD) on the same 50 SEVIR samples and report the averaged results in Fig. 13. We compare the conventional velocity-space prediction strategy (v-pred), the proposed data-space prediction strategy (x-pred), and the full physics-guided version, x-pred+$\mathcal{L}_{phys}$, i.e., PDRF. We also compare the Backbone with and without AMSE under the same evaluation setting. The temporal spectral analysis shows that AMSE reduces the reconstruction error of the deterministic Backbone, confirming that spectral loss adjustment helps preserve predictable temporal scales. Nevertheless, x-pred and x-pred+$\mathcal{L}_{phys}$ achieve lower temporal spectral reconstruction errors than the AMSE-enhanced Backbone and remain closer to the ground-truth spectrum overall. Together with the quantitative results, these observations indicate that AMSE is useful for deterministic models, but it cannot substitute for the

trajectory stabilization and physics-guided generative modeling introduced by PDRF.

# G. Additional qualitative comparisons

Although all main experiments use the standard five-step inference setting, we additionally perform a 20-step rollout visualization on MeteoNet and Shanghai under the same initial noise condition. This extended rollout is used only for qualitative diagnosis, since five-step intermediate states do not always clearly reveal differences between parameterization strategies. We show only the last five inference states, i.e., Steps 16–20, and compare the conventional velocity-space prediction strategy (v-pred) with our data-space prediction strategy (x-pred) and its physics-guided version (x-pred+$\mathcal{L}_{phys}$). For visualization only, we apply PCA to the recorded rollout states and plot their projections on the first two principal components to diagnose trajectory drift. As shown in Figs. 14 and 15, v-pred and x-pred+$\mathcal{L}_{phys}$ follow trajectories closer to the ideal straight-line path from the initial noise state to the target state than v-pred.

Beyond the above rollout diagnostics, Figs. 16, 17, 18, 19, 20, 21, 22, and 23 provide additional visual comparisons on the SEVIR, MeteoNet, Shanghai, and CIKM datasets, respectively. We visualize the input sequences, the corresponding ground-truth future sequences, and the multi-step predictions from a set of representative methods.

Across these examples, the deterministic backbone U-KAN increasingly over-smooths echo morphology as the forecast horizon extends: compact convective structures tend to merge into more regular blobs, and fine-scale gaps/filaments are gradually suppressed. The extrapolation baseline, pyS-TEPS, largely follows the advection of existing echoes and can retain filamentary textures, but it exhibits limited capability in modeling intensity growth/decay and cell interaction; as lead time increases, echoes often weaken or break into fragmented patches.

Among learning-based baselines, DiffCast frequently produces discontinuous, speckled textures with scattered weak false alarms, and it may lose coherent storm shapes at longer lead times. PercpCast typically yields sharper-looking echoes than purely deterministic predictors, but it still shows noticeable artifacts in these figures: weak speckle-like noise and small isolated echoes can appear, and in some cases the prediction becomes overly concentrated into a single dominant band/blob while surrounding light echoes are underrepresented (more evident in the Shanghai cases). AlphaPre, EarthFarseer, and SimVP often regularize the overall morphology and reduce multi-cell variability, resulting in comparatively uniform echo patterns with smoother boundaries.

Overall, PDRF provides predictions that are visually closer to the ground truth in these comparisons, better preserving coherent transport while maintaining more realistic intensity organization (e.g., keeping relatively sharper boundaries and stronger cores without introducing many isolated speckles). Remaining errors are mainly visible at the longest lead times, where slight positional drift and moderate intensity attenuation persist, especially for sparse echoes and rapidly evolving small cells.

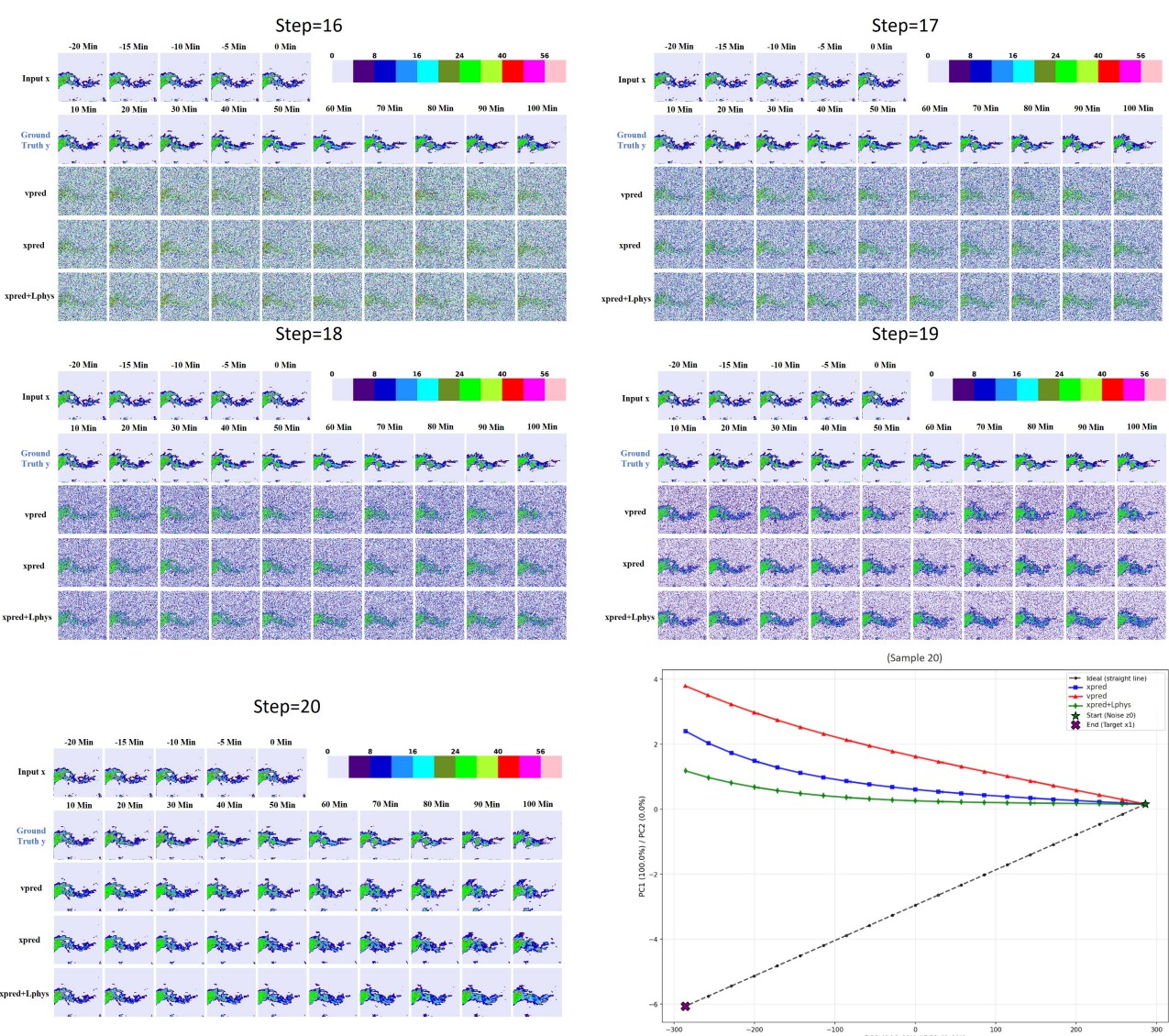

*Figure 14.* 20-step rollout visualization and PCA trajectory analysis on MeteoNet.

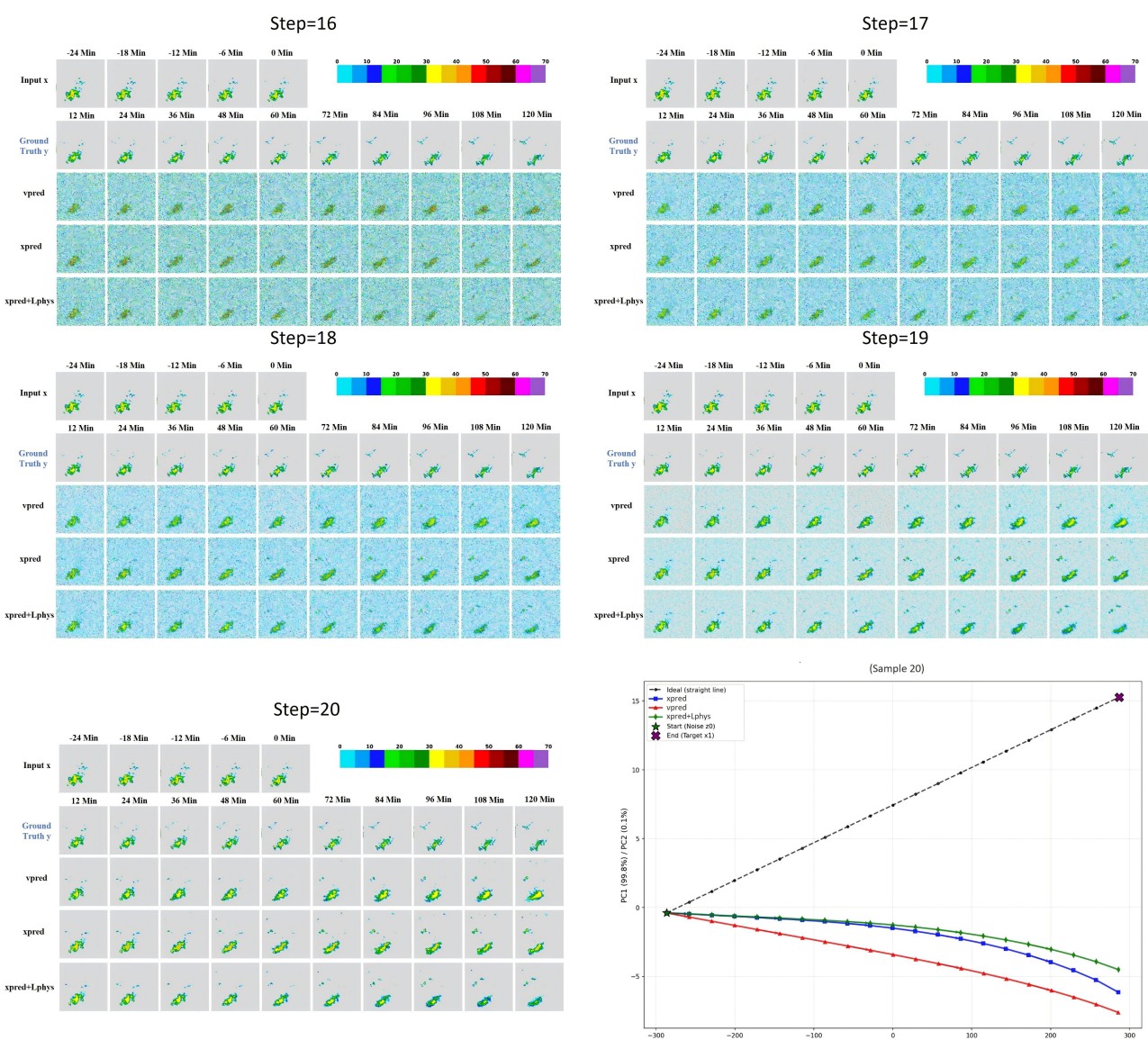

*Figure 15.* 20-step rollout visualization and PCA trajectory analysis on Shanghai.

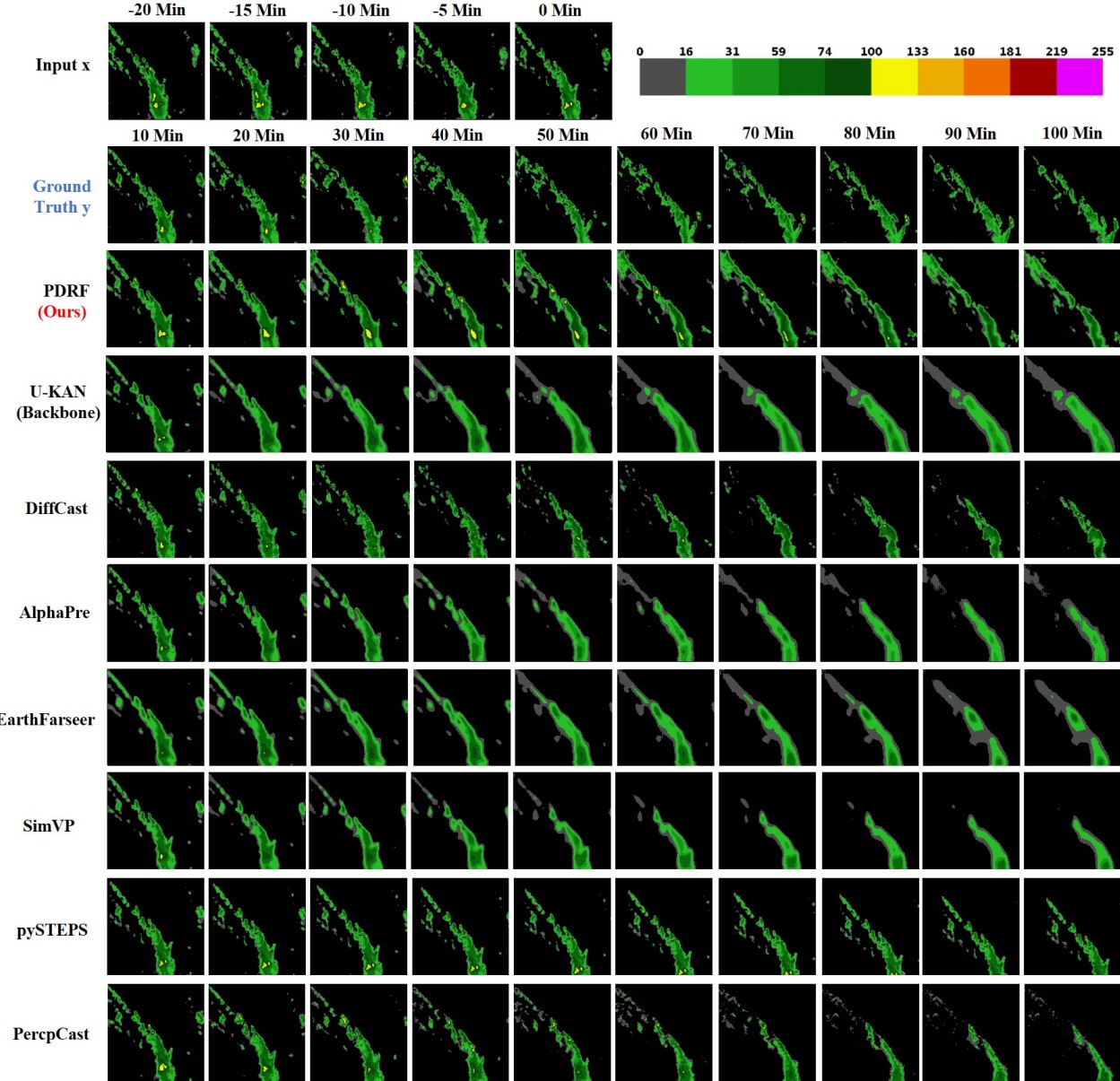

*Figure 16.* **Qualitative comparisons on the SEVIR.**

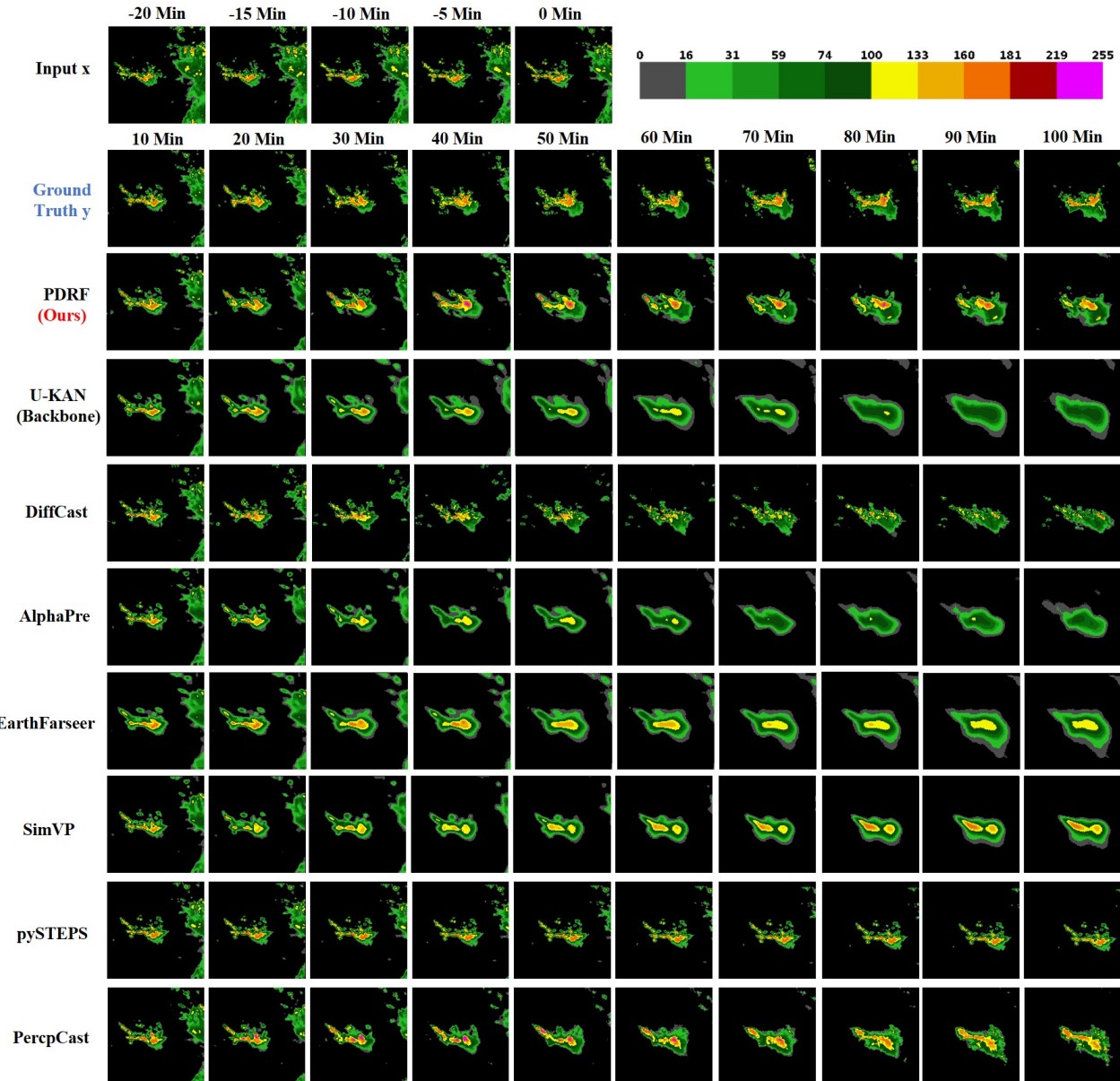

*Figure 17.* **Qualitative comparisons on the SEVIR.**

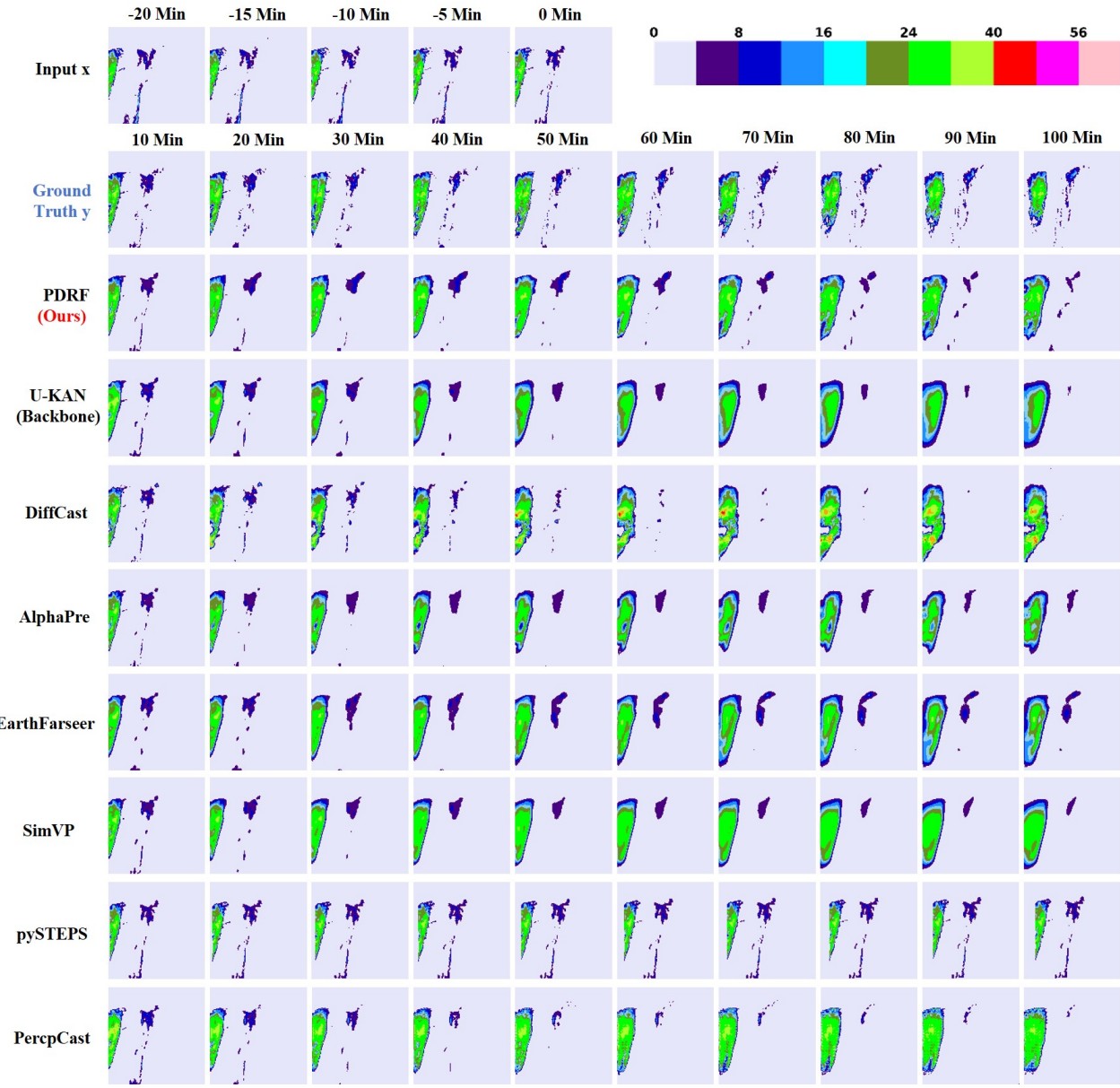

*Figure 18.* **Qualitative comparisons on the MeteoNet.**

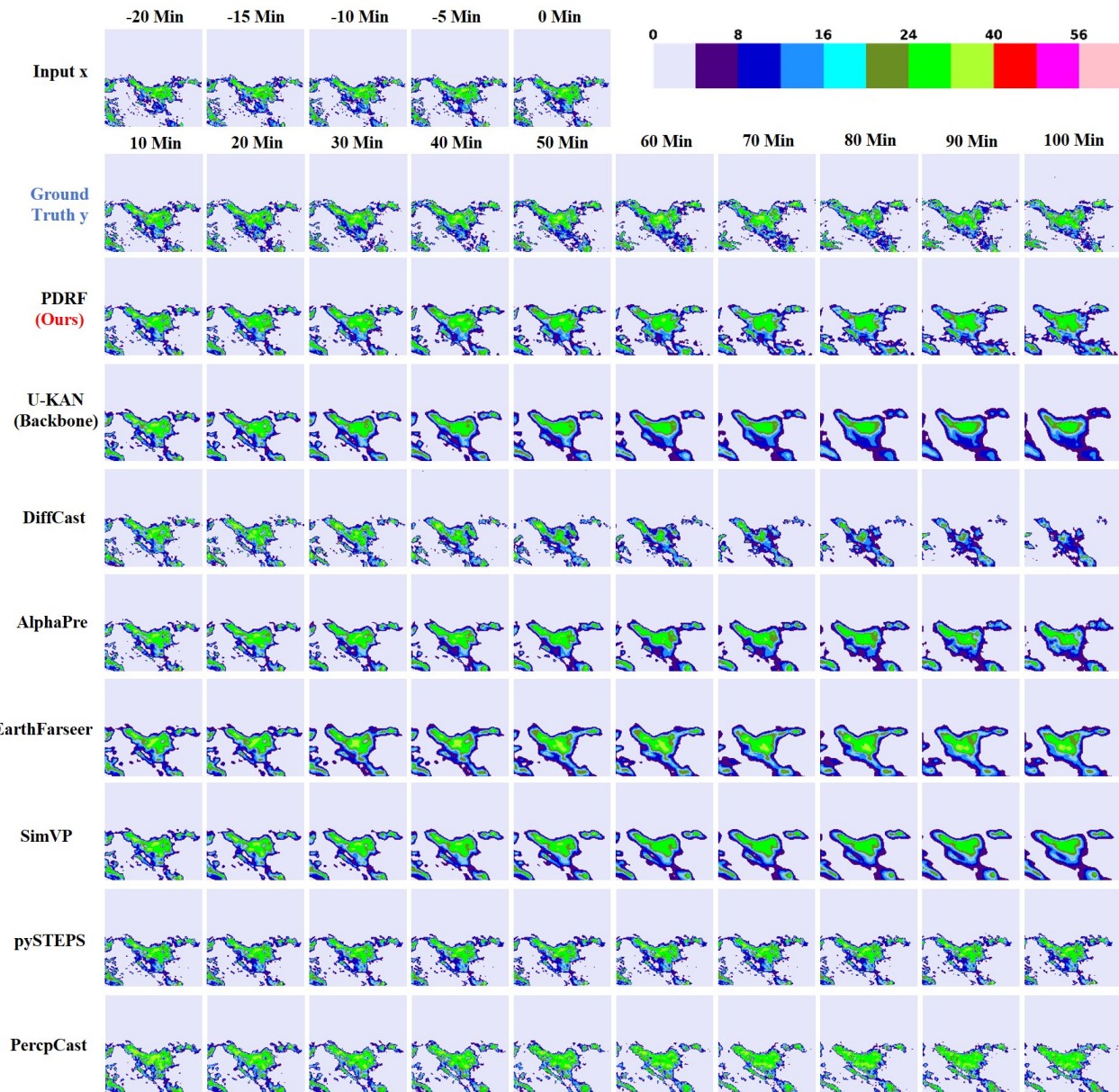

*Figure 19.* **Qualitative comparisons on the MeteoNet.**

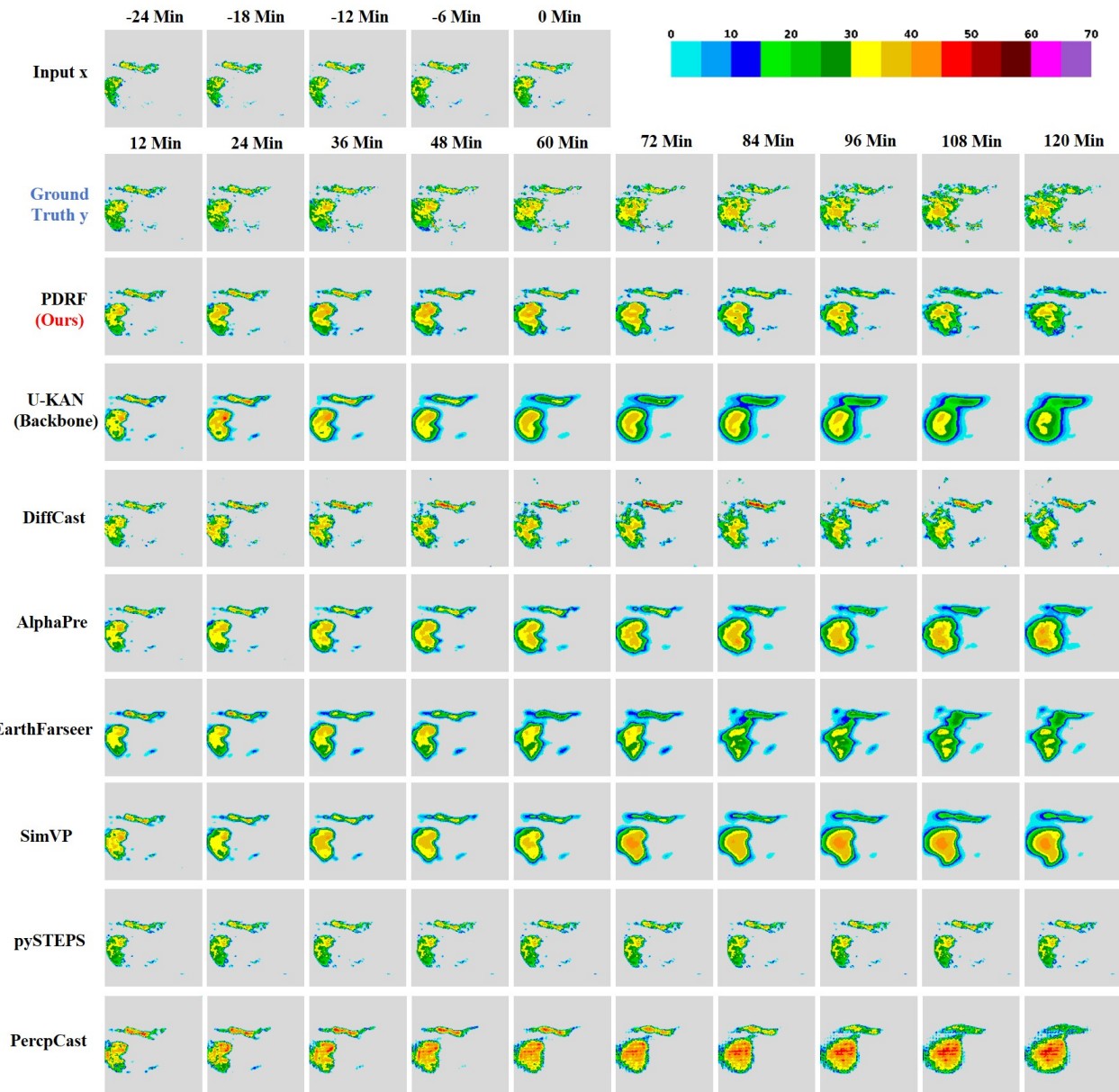

*Figure 20.* **Qualitative comparisons on the Shanghai Radar.**

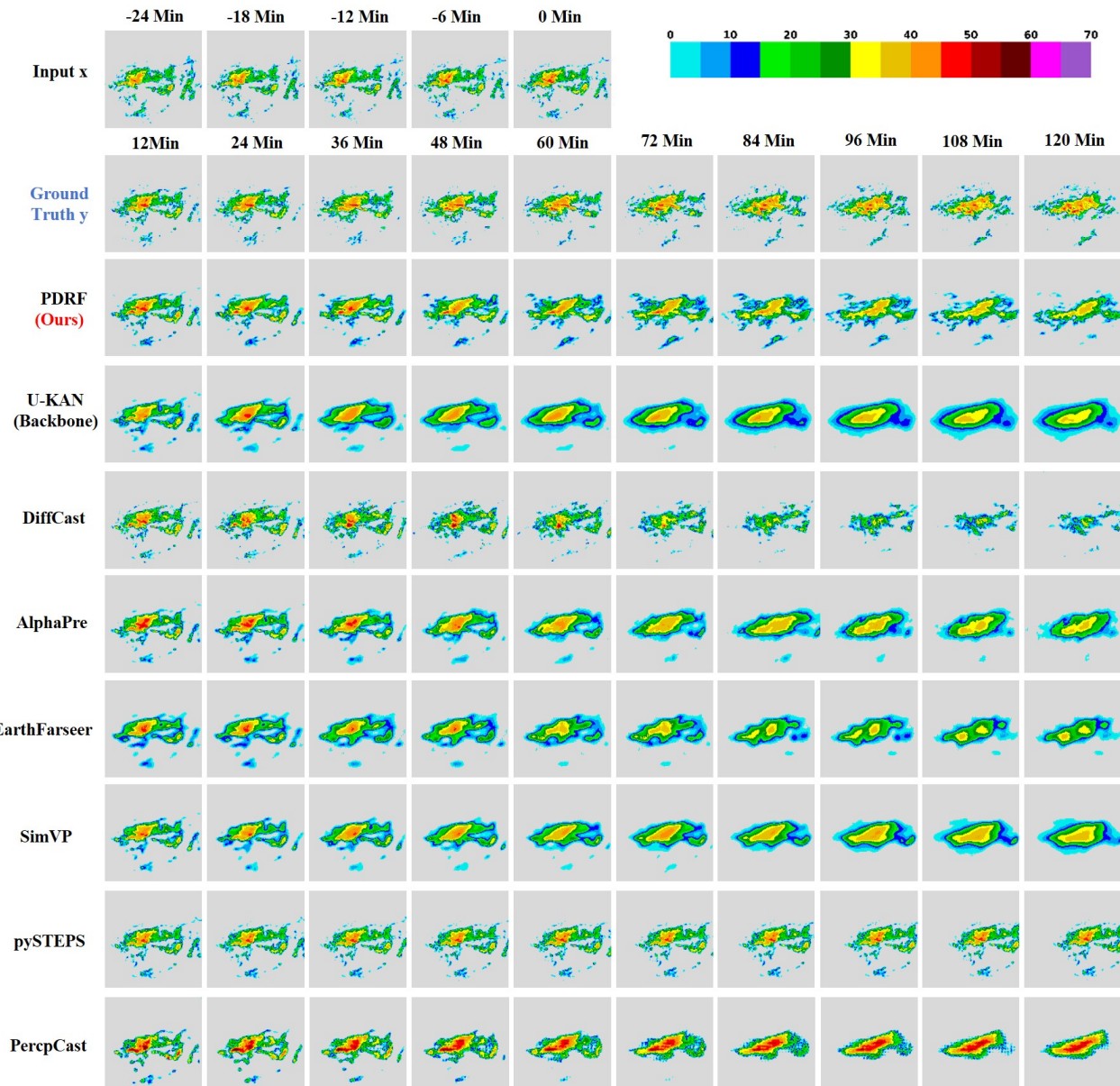

*Figure 21.* **Qualitative comparisons on the Shanghai Radar.**

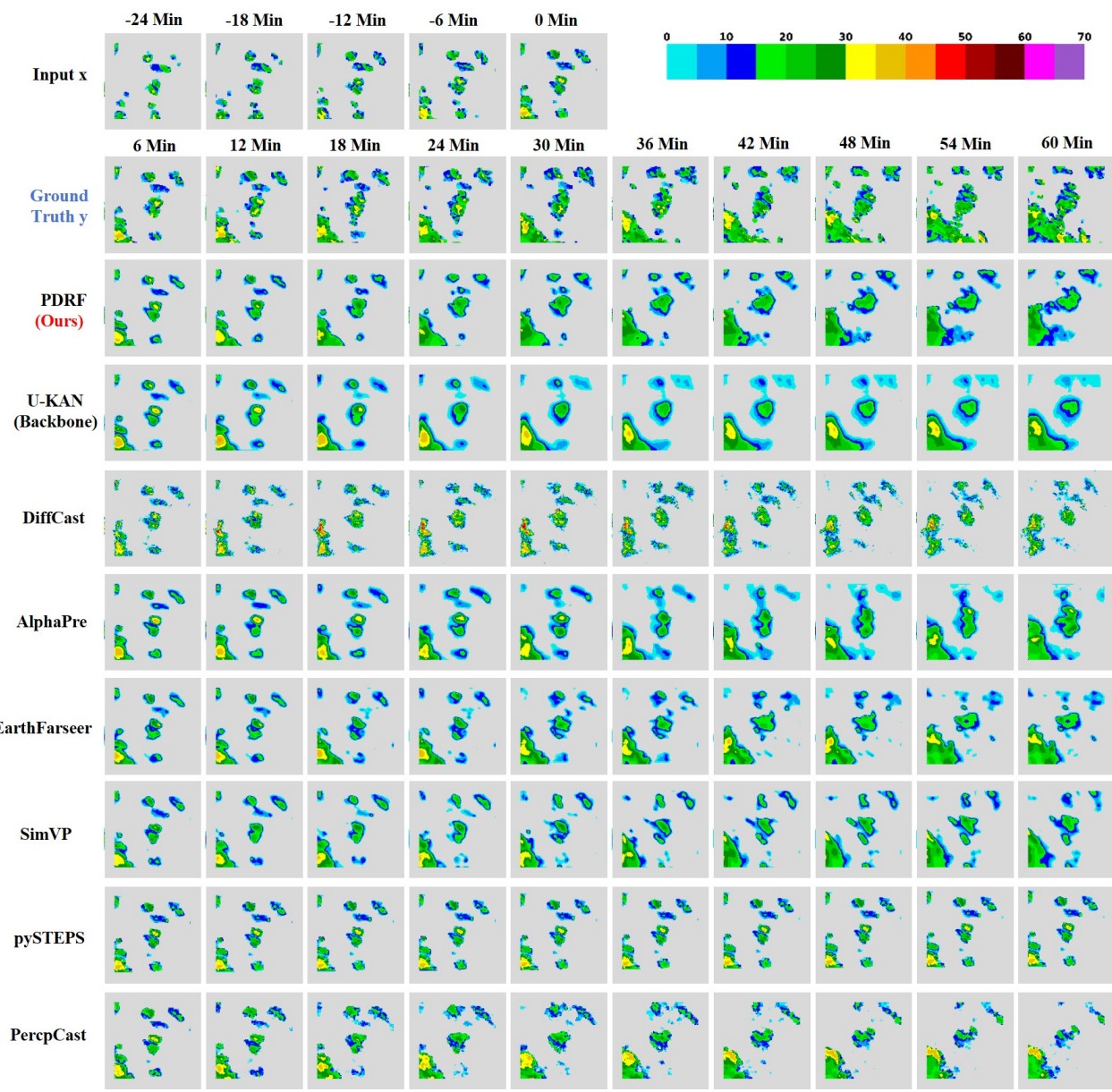

*Figure 22.* **Qualitative comparisons on the CIKM dataset.**

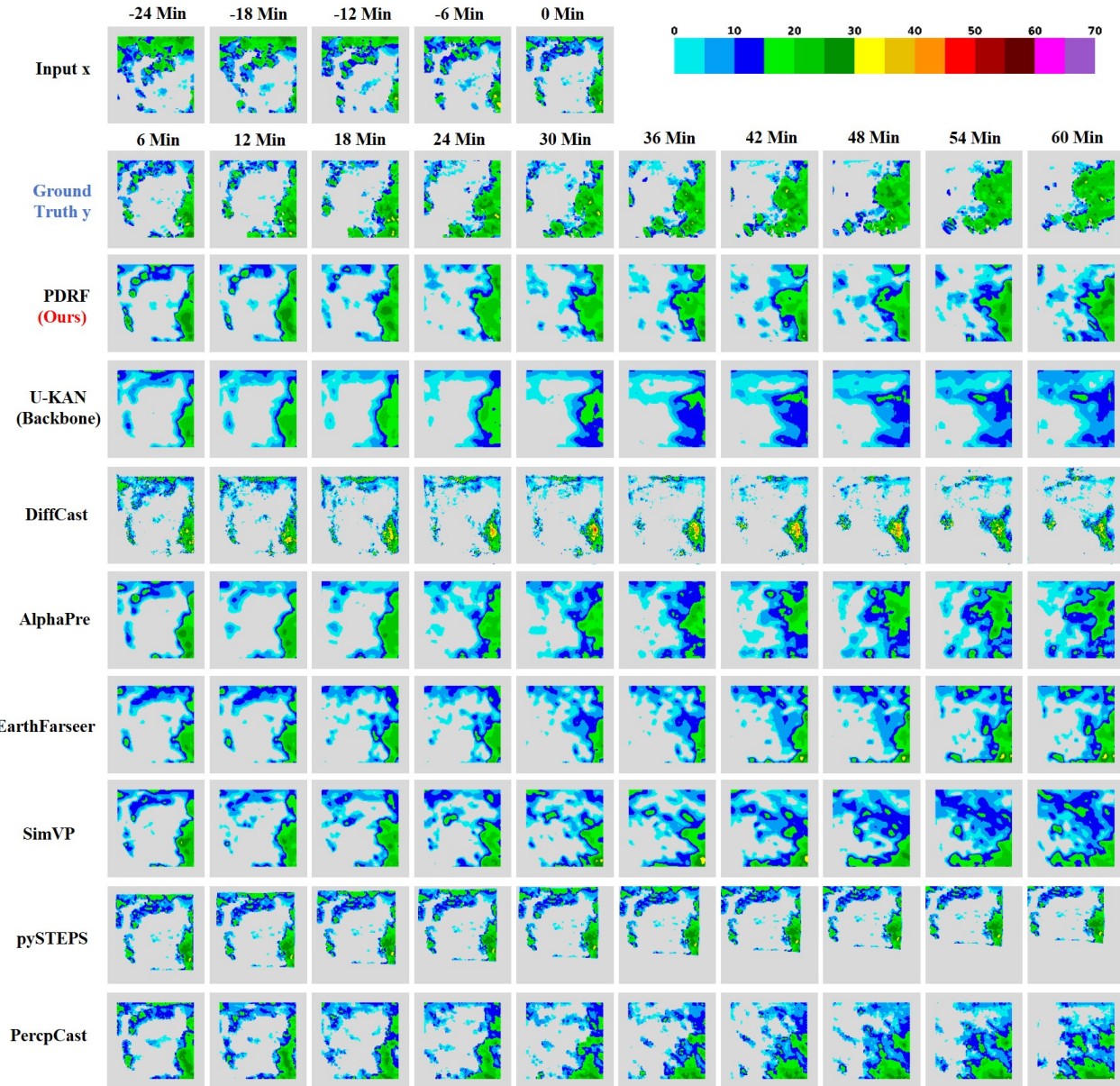

*Figure 23.* **Qualitative comparisons on the CIKM dataset.**

