# OpenReview forum: "Physically-Guided Data-Space Rectified Flow for Precipitation Nowcasting"
_ICML.cc/2026/Conference — ICML 2026 regular_

### Official Review · Reviewer_ZEZp · 2026-03-13

**Soundness:** 3
**Presentation:** 2
**Significance:** 2
**Originality:** 2
**Overall Recommendation:** 4
**Confidence:** 3

**Summary:**

This paper proposes Physically-guided Data-space Rectified Flow (PDRF), a generative framework designed to mitigate "off-manifold drift" in long-horizon precipitation nowcasting. The authors introduce a data-space parameterization that reconfigures the network to predict clean future sequences, analytically inducing an error-correcting restoring force that stabilizes the ODE integration trajectory. Additionally, the model incorporates a physics-guided velocity regularizer based on a Semi-Lagrangian advection prior to maintain large-scale transport coherence while allowing the model to learn complex nonlinear storm evolution from data.

**Compliance With Llm Reviewing Policy:**

Affirmed.

**Final Justification:**

My questions have been addressed by the authors. Therefore, I raise my score.

**Key Questions For Authors:**

1. FAR should also be reported.
2. The physics-guided regularizer relies on the Farneback method for dense motion field estimation. Since this is a classical optical flow technique, how does the model perform in cases of rapid convective initiation or storm decay where pixel-displacement does not represent the underlying growth/decay dynamics?
3. While you assume the Jacobian J_x is contractive along normal directions, could you provide empirical evidence (e.g., via singular value analysis of the Jacobian) that the trained model f_\theta actually behaves as a manifold projector?
4. Precipitation nowcasting often suffers from the "double penalty" problem in pixel-wise losses. While you use a velocity-domain loss ($v$-Space Loss) , did you explore any intensity-weighted loss functions to prioritize the heavy rainfall echoes (e.g., VIL > 160 or dBZ > 40) that are most critical for flood mitigation?

**Limitations:**

1. The model’s physical guidance is heavily predicated on Semi-Lagrangian advection.
2. The framework utilizes the Farneback method for motion estimation.
3. All radar frames are downsampled to a uniform $128\times128$ resolution. This limits the model's ability to forecast micro-scale urban flash floods where sub-kilometer precision is required

**Strengths And Weaknesses:**

Strengths:
1. The submission is well-structured and the narrative is easy to follow.
2. By addressing the accumulation of errors in iterative generative forecasting, the work tackles a critical bottleneck in reliable long-horizon precipitation nowcasting.
3. The work offers a novel combination of generative modeling (Rectified Flow) and classical kinematics (Semi-Lagrangian advection).

Weaknesses:
1. The core architectural components (U-KAN, KAN-VRWKV) and the shift to data-space denoising heavily leverage very recent concurrent works.
2. While the qualitative figures (e.g., Figures 4 and 12) show PDRF's superiority, there is a lack of detailed error analysis for the "Failure Modes" mentioned in the conclusion

---

> ### Author Rebuttal · Authors · 2026-03-29
>
> Thanks for the reviewer's valuable suggestions. We will try to address the reviewer's concerns and are eager to engage in a more detailed discussion with the reviewer.
> ### Q1.
> Thank you for this helpful suggestion. We agree that FAR is an important complementary metric for characterizing false alarms, especially since precipitation nowcasting is typically assessed by multiple competing metrics. In particular, CSI and FAR often trade off against each other, making it difficult for existing models to excel on both. We therefore added FAR together with POD to better reflect the balance between detection and false alarms. Although generative models, including ours, do not always outperform strong deterministic baselines in FAR alone, PDRF achieves a better overall balance than other generative methods by maintaining stronger event-detection ability. This suggestion also highlights an important future direction: introducing preference optimization into precipitation nowcasting to better navigate metric trade-offs and reduce false alarms without sacrificing detection ability.
>
> |Method|SEVIR||MeteoNet||Shanghai||CIKM||
> |-|-|-|-|-|-|-|-|-|
> ||FAR↓|POD↑|FAR↓|POD↑|FAR↓|POD↑|FAR↓|POD↑|
> |PDRF|0.5368|0.5140|0.4064|0.5810|0.4247|0.6082|0.5192|0.4477|
> |DiffCast|0.5407|0.4075|0.5064|0.4240|0.4577|0.4592|0.5792|0.4149|
> |PercpCast|0.5371|0.4978|0.4747|0.4955|0.5069|0.5543|0.5678|0.4439|
> |AlphaPre|0.3437|0.3987|0.3147|0.5109|0.3422|0.5411|0.5030|0.3983|
> |EarthFarseer|0.4701|0.4253|0.3918|0.4833|0.3799|0.4611|0.5031|0.3458|
> |SimVP|0.4122|0.4141|0.3458|0.5228|0.3663|0.5355|0.4890|0.3680|
> ### Q2.
> We thank the reviewer for this important question. We agree that, in cases such as rapid convective initiation or storm decay, pixel displacement estimated by Farneback optical flow is no longer a reliable proxy for the underlying growth/decay dynamics. This is precisely why, in PDRF, the Farneback-based Semi-Lagrangian prior is used only as a soft regularizer that provides coarse transport guidance, rather than as a hard supervision signal for local evolution. As discussed in our response to Reviewer 2, Q2, evaluations on rule-filtered challenging cases still show consistent gains from the physical regularizer. This suggests that, although imperfect in such scenarios, the optical-flow prior remains a useful auxiliary transport prior.
> ### Q3.
> Thank you for this thoughtful suggestion. Directly characterizing the full Jacobian spectrum is computationally infeasible in our high-dimensional spatiotemporal setting, so we instead use perturbation-based directional sensitivity analysis as a tractable empirical proxy. The results show that the trained model is consistently more sensitive to tangent perturbations than to orthogonal normal perturbations, which is consistent with our geometric interpretation that the learned mapping preserves in-manifold variations while being relatively contractive to off-manifold perturbations. While this does not establish that $f_\theta$ is a strict manifold projector, it does provide empirical evidence for projector-like, locally manifold-attracting behavior underlying our restoring-force interpretation. We further validate this behavior on the MeteoNet dataset, where the same directional sensitivity pattern is observed, indicating that the finding is not specific to a single benchmark.
>
> |Perturbation Direction|Mean Response ± SD|Implied Behavior|Consistency|
> |-|-|-|-|
> |Tangent(along estimated manifold) |$9.15\pm4.24$|Structure-preserving($\sigma\approx1$)|50/50|
> |Normal(orthogonal)| $4.28\pm1.72$|Contractive($\sigma<1$) |50/50|
> |Contrast Ratio(Tangent/Normal)| $2.09\pm0.17$|Asymmetric sensitivity|95% CI: [1.70, 2.31]|
>
> **Note:** Statistical Support: Paired $t$-test, $t(49)=13.60$, $p<10^{-17}$, Cohen's $d=1.94$ (very large effect).
> ### Q4.
> Thank you for this valuable suggestion. In our prior work on deterministic models, we observed that intensity-weighted loss can improve heavy-rainfall prediction. In this work, we initially did not include it in order to better isolate the effects of data-pred and physics guidance. Following the reviewer’s suggestion, we added two lightweight experiments during rebuttal: (A) PDRF with intensity-weighted loss (using 20/30/35/40 dBZ thresholds on Shanghai), and (B) Backbone with the same weighting as a control. The results show consistent improvements on large-scale metrics in both settings, suggesting that intensity-aware weighting is a useful complementary design.
>
> |Method|CSI-35↑|CSI-40↑|FAR↓|POD↓|
> |-|-|-|-|-|
> |A|0.4073|0.3044|0.4207|0.6131|
> |B|0.3854|0.2560|0.3802|0.5404|
>
> ### Limitations.
> Thank you for highlighting these limitations. We agree that resizing all radar frames to 128×128 inevitably reduces sensitivity to fine-scale structures, such as urban flash floods. This design choice was made for computational efficiency and stable benchmarking under the current experimental settings. Extending PDRF to higher-resolution inputs is an important direction for future work.

---

> > ### Author Rebuttal · Reviewer_ZEZp · 2026-04-04
> >
> > Thanks for the response. Most of my questions have been answered. I will raise my score.

---

> > > ### Author Response · Authors · 2026-04-04
> > >
> > > Thank you very much for your positive and encouraging feedback. We sincerely appreciate your recognition that our response has adequately addressed your concerns. We are also truly grateful for your insightful comments and clear understanding of this research direction, which have been very valuable in helping us improve the manuscript. Your thoughtful feedback has not only strengthened our work, but also broadened our perspective on this problem. We truly appreciate your recognition and support.

---

### Official Review · Reviewer_KdfD · 2026-03-13

**Soundness:** 3
**Presentation:** 2
**Significance:** 3
**Originality:** 3
**Overall Recommendation:** 4
**Confidence:** 3

**Summary:**

The paper presents a new model for precipitation nowcasting that achieves better performance on important nowcasting metrics on various datasets through several innovations: (1) modified rectified flow formulation where the flow is anchored to a physically plausible manifold through an anlytical velocity field that forces the model to predict clean states (2) a soft regularization that uses an advection guess for the future state solved via semi Lagrangian method and (3) architectural changes in the backbone to ingest these innovations.

**Compliance With Llm Reviewing Policy:**

Affirmed.

**Final Justification:**

The rebuttal has strengthened the paper. Some evaluation details like PSD and AMSE losses probably need a bit more comprehensive evaluation but the authors are committed to a detailed discussion in the revision. I continue to support the paper's acceptance and retain my score.

**Key Questions For Authors:**

- The paper was a little dense to me. The ideas seemed interesting and straightforward but there was a lot of intuition building with heavy terms (ex: geometry stabilization with restoring force or "which preferentially damps the off-manifold (normal) components of perturbations and thereby suppresses off-manifold drift under discretization errors, improving long-horizon stability". I wasn't sure any of the results demonstrate these well enough. Would it make sense to show the evolving flow during inference using CRF, then with the manifold-constrained one and with the SL prior? It would be interesting to see how the model is pushing the noise vector in some physics-informed way towards the generated samples and how it differs from standard flow matching. It would also be helpful to write full forms of the various abbreviations in the results table captions as well since there were too many of them
- Given the chaotic nature of prediction, it might be useful to show probabilistic metrics (CRPS, spread-skill) that would quantify some of the physical realism better. Similarly power spectra as well. If there are some conservation metrics, these would be even better.
- The ablations were nice but it seemed like architecture components were entangled with the very interesting physics-guided RF. It would be useful to see backbone + CRF + data-pred and backbone + CRF + data_pred + L_phys and backbone + CRF + L_phys (with lambda=1). Since the SL guess is not the best (but obeys advection principles), would be interesting to see what the model predicts without appropriate anchoring from the actual target.
- Fig 4 and 5 are too small I think. Making them bigger and more legible would be very helpful
- I am curious as to how spectral loss adjustments to deterministic models fare in this setting. For example: https://openreview.net/pdf?id=YNh77OLRid uses AMSE loss that simply tells the model to fit only the predictable scales and preserve the amplitude in general. Maybe the best deterministic model with AMSE would show if this helps?
- Similarly, models could be trained as ensembles to capture the high resolution (ex: CRPS training: https://arxiv.org/pdf/2506.10772v1). These would be another category of baselines to check.

**Limitations:**

There is no limitations section.

**Strengths And Weaknesses:**

Strengths:
- The RF and SL soft penalty methodologies are physically motivated, constrained the nowcasting results to follow physically consistent paths through flow matching
- The results show superior performance of their method over existing baselines. Several baselines including deterministic ones are compared against
-  Lots of ablations demonstrating impact of individual components

Weaknesses:
- The paper reads a little dense with many terminologies
- Physical consistency arguments are strong motivations in the paper but the results are demonstrated only using image-based metrics

---

> ### Author Rebuttal · Authors · 2026-03-29
>
> We are grateful for the reviewer's acknowledgment of our work and their detailed feedback, which will help us refine our research.
> ### Q1.
> Thank you for this thoughtful and insightful feedback. We fully agree that, for mechanism-driven claims, when the paper contains several terms that are not sufficiently intuitive and may be difficult to follow, it is important to make the intuition directly observable from the results. To address this, we will add an inference-time rollout comparison under the same setup, starting from the same initial noise and comparing standard CRF, the manifold-constrained data-pred variant, and the full model with the Semi-Lagrangian (SL) prior. By visualizing intermediate states and their deviation from the reference RF path, we aim to show how the trajectory evolves and how data-pred and the SL prior affect this process. This also complements the rollout-drift diagnostic discussed in our response to Reviewer 1, Q1. In addition, we will spell out all abbreviations in the result-table captions to improve readability.
> ### Q2.
> Thank you for the helpful suggestion. We agree that probabilistic diagnostics are important for chaotic forecasting, and we have added CRPS in the rebuttal to address this point directly. We also appreciate the suggestions on spread-skill, power spectra, and conservation-related metrics. Due to the limited rebuttal space, we are unable to include all of these analyses here, but we will clarify the discussion in the revision and consider adding further physics-oriented evaluations in the supplementary material.
>
> |Method|SEVIR|MeteoNet|Shanghai|CIKM|
> |-|-|-|-|-|
> |PDRF|0.0360|0.0125|0.0199|0.0388|
> |DiffCast|0.0369|0.0152|0.0214|0.0419|
> |PercpCast|0.0373|0.0136|0.0221|0.0417|
> |AlphaPre|0.0310|0.0120|0.0218|0.0408|
> |EarthFarseer|0.0397|0.0150|0.0233|0.0412|
> |SimVP|0.0389|0.0129|0.0208|0.0397|
>
> ### Q3.
> Thank you for this insightful suggestion. We have added the three requested variants under the same backbone and training setup: (A) backbone + CRF + data-pred, (B) backbone + CRF + data-pred + $L_{\text{phys}}$ ($\lambda=0.1$), and (C) backbone + CRF + $L_{\text{phys}}$ ($\lambda=1$). The results suggest that the SL prior is most effective as an auxiliary physics constraint rather than the main anchor for target generation. This is consistent with Table~3: because $L_{\text{phys}}$ is defined through the same induced-velocity formulation as data-pred, optimizing them in the same derivative space leads to better-aligned gradients, while combining $L_{\text{phys}}$ with velocity prediction introduces a mismatch and hurts performance. Since the SL estimate is only approximate, $L_{\text{phys}}$ is better used with a moderate weight.
>
> |Method|CSI-M↑|CSI-35↑|CSI-40↑|HSS↑|SSIM↑|MSE↓|CRPS↓|
> |-|-|-|-|-|-|-|-|
> |A|0.4241|0.3821|0.2835|0.5600|0.7925|31.76|0.0189|
> |B|0.4335|0.3934|0.2943|0.5652|0.8033|29.63|0.0180|
> |C|0.3906|0.3389|0.2381|0.5002|0.6384|36.15|0.0252|
> ### Q4.
> Thank you for your valuable suggestion. We will adjust Figures 4 and 5 to a larger size for better legibility in the revised manuscript.
> ### Q5.
> Thank you for the helpful suggestion. We ran additional experiments with AMSE on deterministic baselines. Backbone+AMSE (B) shows clear improvements across datasets, suggesting that spectral-loss adjustment can indeed be beneficial in this setting. In contrast, AlphaPre+AMSE (A) yields only limited gains in our experiments.
>
> |SEVIR|B|A|MeteoNet|B|A|
> |-|-|-|-|-|-|
> |CSI-M↑|0.3243|0.3349|CSI-M↑|0.4079|0.3841|
> |CSI-181↑|0.1526|0.1572|CSI-24↑|0.3904|0.3655|
> |CSI-219↑|0.0759|0.0665|CSI-32↑|0.2102|0.1926|
> |HSS↑|0.4161|0.4225|HSS↑|0.5438|0.5123|
> |SSIM↑|0.6121|0.6230|SSIM↑|0.8232|0.7956|
> |MSE↓|401.16|396.33|MSE↓|10.27|11.43|
> |CRPS↓|0.0387|0.0370|CRPS↓|0.0128|0.0141|
>
> |Shanghai|B|A|CIKM|B|A|
> |-|-|-|-|-|-|
> |CSI-M↑|0.4068|0.3478|CSI-M↑ |0.3199|0.2924|
> |CSI-35↑|0.3671|0.3071|CSI-35↑|0.2203|0.1907|
> |CSI-40↑|0.2538|0.1874|CSI-40↑|0.1479|0.1116|
> |HSS↑|0.5384|0.4743| HSS↑|0.4109|0.3755|
> |SSIM↑|0.7755|0.6840| SSIM↑|0.6155|0.5655|
> |MSE↓|28.89|33.49|MSE↓|40.28|38.88|
> |CRPS↓|0.0197|0.0237|CRPS↓|0.0420|0.0437|
>
> ### Q6.
> We thank the reviewer for this valuable suggestion. We implemented a CRPS-based baseline using UKAN (our backbone architecture) with parameter-space perturbations (dim=32). Specifically, we trained $J$=3 independent models with the Fair CRPS loss, each generating $N$=4 ensemble members, resulting in 12 members in total on the Shanghai dataset. We also evaluated $J$=2 for comparison, while scaling to $J$=4 was not feasible under our current hardware setup. Importantly, the core methodology—parameter-space perturbation combined with deep ensembles and CRPS training—remains fully consistent with the referenced approach, with only a reduced scale in implementation.
>
> |Method|CSI-M↑|CSI-35↑|CSI-40↑|HSS↑|SSIM↑|MSE↓|CRPS↓|
> |-|-|-|-|-|-|-|-|
> |$J$=2, $N$=4|0.3722|0.3117|0.1808|0.4895|0.8054|24.21|0.0179|
> | $J$=3, $N$=4 | 0.3612 | 0.3035 | 0.1777 | 0.4810| 0.8012 | 25.30 | 0.0180 |

---

> > ### Author Rebuttal · Reviewer_KdfD · 2026-04-03
> >
> > Thank you very much for the responses. I believe the CRPS scores are a good addition as well as the other ablations. It would be nice to see Q1 visualizations before the rebuttal ends - that should help clarify some of the paper text. I believe anonymous links are allowed in the rebuttal. I hope the authors also consider power spectra in their results. Losses like AMSE seem to be helping your backbone and power spectral density plots can show this well.

---

> > > ### Author Response · Authors · 2026-04-04
> > >
> > > ### Q1
> > > We sincerely thank the reviewer for this insightful suggestion. In response to Q1, we have added additional visualization results to the rebuttal. Specifically, on both the Shanghai and Meteo datasets, we conduct controlled case studies under identical initial noise conditions to compare v-pred, x-pred, and $L_{\text{phys}}$ in a fair and consistent manner.
> > >
> > > The added visualizations include:
> > > (a) trajectory drift plots based on PCA projections, showing that v-pred exhibits more pronounced deviation, while x-pred and $L_{\text{phys}}$ tend to follow trajectories closer to the ideal straight path; and
> > > (b) step-by-step sampling visualizations, which illustrate the frame-by-frame evolution throughout the sampling process.
> > >
> > > Since our paper adopts 5-step inference as the standard setting, all main experiments are consistently conducted under this setup. However, when directly visualizing intermediate states under 5-step inference, the differences between methods are not always sufficiently discernible. To provide a clearer qualitative comparison, we therefore additionally include a 20-step inference visualization and present only its last five steps, which offer a more fine-grained view of the late-stage sampling behavior. This supplementary visualization is introduced solely to facilitate clearer observation and comparison; all quantitative results and the main experimental findings in the paper remain based on the unified 5-step inference setting.
> > >
> > > For fairness, all comparisons are conducted under the same input conditions and generation targets. Detailed results are available at https://anonymous.4open.science/r/Anonymize1-B4B1/. We sincerely appreciate this helpful suggestion, as these additional visualizations further clarify the differences among v-pred, x-pred, and $L_{\text{phys}}$.
> > >
> > > ### Q2
> > > We sincerely thank the reviewer for the valuable suggestion to examine power spectral density (PSD). Following your advice, we conducted preliminary frequency-domain analyses by averaging PSD results over 50 samples for each setting. Specifically, on the SEVIR dataset, we compare the PSD of v-pred, x-pred, and L_phys. In addition, on the CIKM and MeteoNet datasets, we compare the PSD of the backbone with and without the proposed AMSE loss. The detailed results are available at [https://anonymous.4open.science/r/Anonymize1-B4B1/]. While these findings are still preliminary, we will further expand this analysis and provide a more comprehensive discussion in the revised manuscript. We will also continue to explore PSD analysis in future research.
> > >
> > > We are sincerely grateful for every suggestion from the reviewer, as these thoughtful comments have not only helped improve our manuscript but also broadened our perspective on this work. If there are any further questions or concerns, we will be glad to respond actively and clarify them to the best of our ability.

---

### Official Review · Reviewer_ZrZa · 2026-03-13

**Soundness:** 3
**Presentation:** 4
**Significance:** 3
**Originality:** 3
**Overall Recommendation:** 4
**Confidence:** 4

**Summary:**

This work proposes a physically-guided data-space rectified flow framework PDRF for rectified flow in long-horizon precipitation nowcasting. The proposed model analytically induces a vector field by directly predicting clean future sequences, leveraging an implicit restoring force to suppress off-manifold drift. For physical guidance, it adopts a Semi-Lagrangian advection prior as a soft constraint to ensure large-scale transport coherence while retaining the ability to learn local nonlinear evolution. Additionally, it integrates modules such as FCM, CGSTF, and WGSC to adapt to the spatiotemporal characteristics of radar echoes.

**Compliance With Llm Reviewing Policy:**

Affirmed.

**Final Justification:**

I appreciate the extensive efforts the authors have made in the rebuttal. These efforts have addressed some of my concerns and enhanced the rationality of the paper. However, the limitation identified in Q3 prevents me from raising the score further.

**Key Questions For Authors:**

1. This work utilizes a Semi-Lagrangian advection prior to ensure large-scale transport consistency, but has the proposed model investigated the degree of alignment between this prior and the physical laws governing precipitation? The Semi-Lagrangian method is suitable for passive scalar transport, yet precipitation evolution involves nonlinear physical processes such as water vapor phase transition and convective development. Could simplistic advection constraint conflict with real-world physical mechanisms?
2. The Semi-Lagrangian advection prior relies on Farneback optical flow to estimate historical motion fields, which can only capture translational motion. Does it achieve satisfactory performance when modeling complex dynamics of precipitation systems, such as deformation, merging, splitting, or the explosive growth of convective cells? Can the model maintain the physical consistency of long-horizon forecasts when handling complex precipitation events like severe convection?
3. Compared with methods that integrate fluid dynamics equations with physical priors to constrain the generation process, does the simplified advection constraint limit the physical expressive capacity of the proposed model?

**Limitations:**

This work lacks an analysis of the model's limitation regarding application scenarios and other potential aspects.

**Strengths And Weaknesses:**

The proposed model effectively mitigates the limitation of off-manifold drift in traditional physical forecasting models under practical application scenarios, addressing long-horizon distortion from both geometric stability and physical rationality perspectives. Meanwhile, the manuscript provides valid derivations and analyses of the framework’s operational mechanisms: modules including FCM, CGSTF, and WGSC in the backbone network tackle specific challenges in precipitation nowcasting—such as cross-scale information interaction, spatiotemporal misalignment, and high-frequency detail preservation—rather than simply reusing generic architectures. Comprehensive experiments further validate the proposed model’s performance advantages over various types of off-the-shelf methods in the corresponding dataset scenarios. The weaknesses parts of the proposed model is listed in the Key Questions For Authors section.

---

> ### Author Rebuttal · Authors · 2026-03-29
>
> We are grateful for the reviewer's acknowledgment of our work and their detailed feedback, which will help us refine our research.
> ### Q1.
> Thank you for this important and well-posed question. We agree that a Semi-Lagrangian advection prior is only physically consistent with the transport component of precipitation dynamics, and does not capture nonlinear processes such as phase transitions or convective development. In this work, we do not attempt to establish a strict PDE-level alignment between the prior and the full physical laws governing precipitation. Instead, we assess their compatibility empirically: the SL prior is introduced as a soft constraint in the induced-velocity space, and its alignment with real dynamics is reflected in whether it improves predictive performance without degrading nonlinear structures. Our ablations (Table~3 and Reviewer 3, Q3) show that a moderate SL weight consistently improves performance, whereas overly strong regularization leads to degradation, indicating that the SL prior is partially aligned with the transport component of precipitation dynamics but may conflict with true dynamics when over-constrained. This suggests that the prior captures a physically meaningful but incomplete component of precipitation evolution.
>
> ### Q2.
> We thank the reviewer for this important question. We agree that Farneback optical flow provides only a coarse approximation of motion and is limited in representing complex precipitation evolution. In PDRF, however, the Semi-Lagrangian prior is not intended to model these dynamics by itself; it serves only as a soft transport prior that provides coarse transport guidance, while the nonlinear local evolution is primarily captured by the data-driven component. To directly assess this issue, we further evaluated PDRF on rule-filtered complex subsets from both MeteoNet and Shanghai, where the subsets were heuristically constructed using intensity- and morphology-based criteria to capture challenging non-translational evolution patterns. We observed consistent gains from the physical regularizer. This suggests that, although imperfect for complex non-translational dynamics, the Farneback-based prior remains beneficial on these challenging cases.
>
> |Dataset|Category|Metric|λ=0.1|λ=0|
> |-|-|-|-|-|
> |MeteoNet|Rule-filtered Complex Cases (272)|CSI-M↑|0.4094|0.3935|
> |||CSI-32↑|0.1989|0.1812|
> |||HSS↑|0.5495|0.5311|
> |||MSE↓|33.26|40.36|
> ||Splitting-like Cases (18)|CSI-M↑|0.4310|0.4090|
> ||Merging-like Cases (18)|CSI-M↑|0.4288|0.4061|
> ||Severe-Convection-like Cases (251)|CSI-M↑|0.4078|0.3928|
> |||CSI-32↑|0.2015|0.1840|
> |Shanghai|Rule-filtered Complex Cases (217)|CSI-M↑|0.3674|0.3569|
> |||CSI-40↑|0.2015|0.1905|
> |||HSS↑|0.5059|0.4930|
> |||MSE↓|61.78|72.67|
> ||Splitting-like Cases (63)|CSI-M↑|0.4162|0.4044|
> ||Merging-like Cases (77)|CSI-M↑|0.3788|0.3672|
> ||Severe-Convection-like Cases (103)|CSI-M↑|0.3383|0.3303|
> |||CSI-40↑|0.1852|0.1774|
>
> **Note:** The numbers in parentheses denote the sample counts of the corresponding subsets.
> ### Q3.
> Thank you for this insightful question. We agree that, compared with approaches that incorporate fuller fluid-dynamical constraints, a simplified advection prior is inherently less expressive in representing the full physics of precipitation evolution. In particular, processes such as moisture convergence, phase change, and convective instability cannot be captured by advection alone. However, in our current setting the model operates only on radar echo sequences, without access to multimodal atmospheric variables such as wind, humidity, temperature, or pressure, which makes it difficult to impose stronger physics constraints in a consistent way. Therefore, we adopt the Semi-Lagrangian prior as a lightweight and robust transport guidance rather than a full physical model. In this sense, the simplified advection constraint does limit physical expressive capacity to some extent, but it is a practical compromise under the current data modality. An important future direction is to incorporate richer atmospheric variables and stronger dynamical constraints, such as moisture continuity or simplified Navier–Stokes formulations, so that both transport-dominated and strongly nonlinear convective processes can be better modeled.
>
> ### Limitations.
> Thank you for pointing this out. We agree that the current work mainly validates the proposed framework under a unified setting on public nowcasting benchmarks, while its applicability across different scenarios still requires further study. In particular, we have not yet systematically examined its behavior under different weather regimes or varying reliability of the physical prior. More broadly, since the physics-guided term is introduced as an auxiliary constraint rather than tied to a specific dynamical form, an important future direction is to incorporate richer physics priors and adapt the guidance more flexibly to different weather conditions.

---

> > ### Author Rebuttal · Reviewer_ZrZa · 2026-04-04
> >
> > I appreciate the authors for their response. My concerns regarding Q1 and Q2 have been  addressed. I have noted the authors' response to Q3, and the content therein may be expanded and incorporated into the Limitations and Future Work section.

---

> > > ### Author Response · Authors · 2026-04-04
> > >
> > > Thank you for your careful and constructive feedback, and we also appreciate your recognition that our responses to Q1 and Q2 have addressed those concerns. Your comments on Q3 are very insightful and helpful in clarifying the current scope and limitations of our work. Following your suggestion, we will further expand and revise the discussion related to Q3 in the revised manuscript, and explicitly incorporate it into the “Limitations and Future Work” section to better articulate the current limitation and promising directions for future investigation. If you have any further questions or suggestions, we would be truly grateful for your continued feedback, as your comments are very helpful in improving our manuscript.
> > >
> > > ### Limitations and Future Work
> > > Although the Semi-Lagrangian advection prior introduced in this work empirically helps improve large-scale transport consistency and enhances long-horizon generation stability, its physical role is mainly limited to a coarse constraint on the transport component of precipitation evolution. It should therefore be understood as a lightweight soft kinematic regularizer, rather than a physical model that is strictly aligned with the full governing physics of precipitation. In particular, it cannot explicitly represent complex processes such as phase transitions, latent heat release, strongly nonlinear convective development, or the deformation, merging, and splitting of precipitation systems; when the regularization is too strong, it may also conflict with the true underlying dynamics. At the same time, this design choice is closely related to the current radar-only setting, in which auxiliary atmospheric variables such as wind, humidity, temperature, and pressure are unavailable, making it difficult to impose stronger physical constraints in a consistent manner. A promising direction for future work is to extend the framework toward multimodal meteorological forecasting by incorporating richer atmospheric state variables and stronger dynamical constraints, so that both transport-dominated evolution and highly nonlinear convective processes can be better modeled.

---

### Official Review · Reviewer_nHAL · 2026-03-13

**Soundness:** 3
**Presentation:** 2
**Significance:** 3
**Originality:** 3
**Overall Recommendation:** 4
**Confidence:** 4

**Summary:**

The paper proposes PDRF - a new approach for precipitation nowcasting. The core idea is to replace direct velocity prediction in conditional rectified flow with data-space prediction of the clean future sequence, which then analytically induces the vector field and is argued to suppress off-manifold drift during ODE integration. The method also adds a Semi-Lagrangian velocity regularizer built from a Farneback optical-flow prior, and combines these ideas with a specialized backbone including CGSTF, WGSC, FCM, and KAN-VRWKV blocks. Experiments show strong gains in CSI/HSS and good structural fidelity.

**Compliance With Llm Reviewing Policy:**

Affirmed.

**Final Justification:**

I appreciate the great amount of work done by the authors and probably the paper looks a little bit overcomplicated now. Probably a good addition would be to add also the estimates of the standard deviations of the obtained metrics in order to understand the statistical significance of the obtained gains.

**Key Questions For Authors:**

1.	Can the authors provide a direct diagnostic of drift accumulation during rollout?
2.	Can the authors provide a controlled comparison where the backbone, training recipe, and sampler are fixed, and only velocity prediction versus data-space prediction is changed in the final model?
3.	Are the KAN-VRWKV blocks so significant? Can they be replaced by something simpler?
4.	How sensitive is the method to errors in the Farneback flow used to build the Semi-Lagrangian teacher, especially on difficult cases with strong growth or deformation?

**Limitations:**

The limitations discussion might be improved. The paper should discuss that the stability argument is partly heuristic, that the final gains are not fully disentangled from backbone engineering, and that the physics-guided prior may be unreliable when optical flow is inaccurate or the storm evolution is strongly non-advective.

**Strengths And Weaknesses:**

Strengths. The paper addresses an important problem of maintaining stability and morphology in radar nowcasting. The proposed data-space parameterization is a meaningful modeling choice, and the empirical results show that the approach is practically promising. PDRF shows good or best CSI/HSS metrics across all four datasets, and it is also relatively efficient in computational speed.

Weaknesses. The paper’s strongest claim is about why the method works, but the evidence is not yet strong enough. The stability argument relies on the denoiser acting like a manifold projector and on the resulting Jacobian structure damping off-manifold perturbations, but this is not directly validated in experiments. At present, the paper shows improved forecasting metrics, not direct evidence that off-manifold drift is actually reduced in the sense claimed by the theory.
A second issue is attribution. The final system combines at least three substantial changes at once: data-space parameterization, physics-guided regularization, and a heavily engineered backbone with CGSTF, WGSC, FCM, and KAN-VRWKV. This makes it difficult to determine how much of the final gain should be credited to the central methodological contribution rather than to architecture design. The paper does include ablations, but they are not structured enough to allow one to draw clear conclusions.
A third issue is that the physics prior is limited. The Semi-Lagrangian teacher uses a single Farneback flow field estimated from the last two observed frames and then extrapolates the full horizon from that field. This is a reasonable heuristic, but it is also exactly the kind of prior that can break under strong growth, decay, or deformation. The paper does not really study sensitivity to poor motion estimates.

---

> ### Author Rebuttal · Authors · 2026-03-29
>
> Thanks for the reviewer's valuable suggestions. We will try to address the reviewer's concerns and are eager to engage in a more detailed discussion with the reviewer.
>
> ### Q1.
> Thank you for the suggestion. We added a direct trajectory-level rollout-drift diagnostic under the same backbone, training setup, and fixed 5-step ODE sampler, to compare v-pred and x-pred. For each held-out sample, both methods start from the same initial noise, and we record the intermediate rollout states to measure the normalized off-path deviation $D\_\perp(t)$ from the ideal RF path. Each entry reports v-pred/x-pred (relative reduction), where lower is better.
>
> |Dataset|$D\_\perp(0.2)$|$D\_\perp(0.4)$|$D\_\perp(0.6)$|$D\_\perp(0.8)$|$D\_\perp(1.0)$|MeanOffPath|
> |-|-|-|-|-|-|-|
> |MeteoNet |0.007119/0.006665 (6.38%) |0.014080/0.013345 (5.22%) |0.021007/0.020085 (4.39%) |0.028247/0.027171 (3.81%) | 0.036962/0.035384 (4.27%)|0.021483/0.020530 (4.44%) |
> |Shanghai|0.013096/0.010592 (19.13%)|0.025289/0.021334 (15.64%) |0.037079/0.032467 (12.44%) |0.049083/0.044560 (9.21%) |0.062470/0.058547 (6.28%)|0.037403/0.033500 (10.44%)|
> |SEVIR|0.013073/0.012761 (2.39%)| 0.026146/0.025700 (1.70%) |0.039884/0.039505 (0.95%) | 0.054967/0.054568 (0.73%) | 0.071360/0.070744 (0.87%)| 0.041086/0.040656 (1.05%) |
> |CIKM|0.015148/0.014436 (4.70%)| 0.030151/0.028904 (4.14%)| 0.045355/0.043434 (4.24%)|0.061296/0.058246 (4.98%) | 0.079080/0.073727 (6.77%)| 0.046206/0.043749 (5.32%) |
>
>
> ### Q2.
> Thank you for this important question. We agree that the cleanest comparison is to fix the backbone, training recipe, and sampler, and change only velocity-space prediction versus x-space prediction. Table 2 is designed exactly under this logic: after progressively adding CRF, CGSTF, WGSC, FCM, and VRWKV to the same U-KAN backbone, the resulting model is still the velocity-pred version; the subsequent Data-Pred row changes only the prediction parameterization to x-space, while keeping the rest of the architecture and training pipeline unchanged. This comparison is intentionally conducted without $\mathcal{L}\_{\text{phy}}$. Meanwhile, Table 3 provides a complementary comparison with $\mathcal{L}\_{\text{phy}}$ enabled (Effect of Parameterization, $\lambda = 0.1$), again under matched architecture and training settings, comparing only velocity-pred and data-pred. In addition, Appendix Table 9 further supports this point through the independent PercpCast-v vs. PercpCast-x ablation under identical training and sampling settings. We also additionally include a comparison under the same training environment: the table below presents the results of backbone + CRF + data-pred(B) versus backbone + CRF(A) on the Shanghai dataset.
>
> |Method|CSI-M↑|CSI-35↑|CSI-40↑|HSS↑|SSIM↑|MSE↓|
> |-|-|-|-|-|-|-|
> |A| 0.4087|0.3734|0.2687|0.5345|0.6420|34.78|
> |B| 0.4241|0.3821|0.2835|0.5600|0.7925|31.76|
>
> ### Q3.
> Thank you for the comment. KAN-VRWKV is helpful, but it is not indispensable to our main claim. Its role is to enhance long-range spatiotemporal context modeling at deeper stages of the backbone. As shown in our appendix ablation (Table 7), replacing the default serial VRWKV with linear attention or SE attention yields comparable MSE but consistently weaker CSI/HSS, especially at medium and high thresholds. This suggests that simpler alternatives are feasible, albeit slightly less effective. More importantly, Table 2 shows that the most significant structural recovery comes from switching to Data-Pred, indicating that the core methodological gain stems from the data-space parameterization, while VRWKV should be viewed as a useful architectural enhancement.
> ### Q4 & Limitations.
> We thank the reviewer for raising this important point. The method does exhibit some sensitivity to errors in the Farneback flow. However, the Semi-Lagrangian prior in our framework is only a soft auxiliary constraint rather than a dominant supervision signal. The primary learning signal comes from the data-pred objective, allowing the model to rely on data-driven corrections when the estimated motion is inaccurate. This design mitigates the impact of imperfect optical flow, as the model is not forced to strictly follow potentially erroneous advection fields.
>
> Regarding the limitations, we agree that the stability argument is partly heuristic and that the final performance gains are not fully disentangled from backbone design. To address this, we have included additional ablation studies in our response to Reviewer 3 (Q3), where various architectural components are removed. The results consistently show that improvements from data-pred and the $\mathcal{L}_{\text{phys}}$ loss remain observable even without these modules, indicating that the core gains are not solely dependent on backbone engineering. At the same time, improving the robustness of the physical prior (e.g., through better motion estimation or adaptive weighting under different regimes) remains an important direction to further strengthen the method.

---

> > ### Author Rebuttal · Reviewer_nHAL · 2026-04-04
> >
> > My concerns are more or less resolved, so I increase my score to weak acceptance (4). I appreciate the great amount of work done by the authors and probably the paper will look a little bit overcomplicated now. Probably a good addition would be to add also the estimates of the standard deviations of the obtained metrics in order to understand the statistical significance of the obtained gains.

---

> > > ### Author Response · Authors · 2026-04-04
> > >
> > > Thank you very much for your careful review, professional assessment, and constructive suggestions. We sincerely appreciate your thoughtful feedback, and we are grateful that our revision has addressed your concerns to a large extent.
> > >
> > > We also thank you for your helpful suggestion on reporting the standard deviations of the obtained metrics. We understood your comment as referring to the rollout-drift results in Q1. Due to the response length limit, the corresponding standard deviations were indeed omitted from our previous response. In response, we have now added the corresponding standard deviations for the reported $D\_\perp(t)$ and MeanOffPath results, so as to better reflect the variability of these measurements and provide additional support for interpreting the observed improvements.
> > >
> > > Thank you again for your thoughtful comments and valuable suggestions.
> > >
> > >
> > > | Dataset   | Method      | $D_\perp(0.2)$       | $D_\perp(0.4)$       | $D_\perp(0.6)$       | $D_\perp(0.8)$       | $D_\perp(1.0)$       | MeanOffPath          |
> > > |-----------|-------------|----------------------|----------------------|----------------------|----------------------|----------------------|----------------------|
> > > | MeteoNet  | v-pred      | 0.007119 ± 0.001861 | 0.014080 ± 0.003764 | 0.021007 ± 0.005735 | 0.028247 ± 0.007877 | 0.036962 ± 0.010417 | 0.021483 ± 0.005927 |
> > > |           | x-pred      | 0.006665 ± 0.001984 | 0.013345 ± 0.003981 | 0.020085 ± 0.005998 | 0.027171 ± 0.008105 | 0.035384 ± 0.010429 | 0.020530 ± 0.006097 |
> > > |           | Rel.Red.(%) | 6.38% ↓             | 5.22% ↓             | 4.39% ↓             | 3.81% ↓             | 4.27% ↓             | 4.44% ↓             |
> > > | Shanghai  | v-pred      | 0.013096 ± 0.002320 | 0.025289 ± 0.004802 | 0.037079 ± 0.007597 | 0.049083 ± 0.010818 | 0.062470 ± 0.014527 | 0.037403 ± 0.008002 |
> > > |           | x-pred      | 0.010592 ± 0.002688 | 0.021334 ± 0.005393 | 0.032467 ± 0.008174 | 0.044560 ± 0.011071 | 0.058547 ± 0.014170 | 0.033500 ± 0.008287 |
> > > |           | Rel.Red.(%) | 19.13% ↓            | 15.64% ↓            | 12.44% ↓            | 9.21% ↓             | 6.28% ↓             | 10.44% ↓            |
> > > | SEVIR     | v-pred      | 0.013073 ± 0.006410 | 0.026146 ± 0.013141 | 0.039884 ± 0.020418 | 0.054967 ± 0.030096 | 0.071360 ± 0.039223 | 0.041086 ± 0.021193 |
> > > |           | x-pred      | 0.012761 ± 0.006953 | 0.025700 ± 0.014084 | 0.039505 ± 0.021750 | 0.054568 ± 0.028481 | 0.070744 ± 0.037553 | 0.040656 ± 0.022413 |
> > > |           | Rel.Red.(%) | 2.39% ↓             | 1.70% ↓             | 0.95% ↓             | 0.73% ↓             | 0.87% ↓             | 1.05% ↓             |
> > > | CIKM      | v-pred      | 0.015148 ± 0.003322 | 0.030151 ± 0.006681 | 0.045355 ± 0.010110 | 0.061296 ± 0.013650 | 0.079080 ± 0.017335 | 0.046206 ± 0.010203 |
> > > |           | x-pred      | 0.014436 ± 0.003229 | 0.028904 ± 0.006463 | 0.043434 ± 0.009708 | 0.058246 ± 0.012955 | 0.073727 ± 0.016194 | 0.043749 ± 0.009696 |
> > > |           | Rel.Red.(%) | 4.70% ↓             | 4.14% ↓             | 4.24% ↓             | 4.98% ↓             | 6.77% ↓             | 5.32% ↓             |

---

### Decision · Program_Chairs · 2026-04-30

**Decision:**

Accept (regular)

**Comment:**

The paper introduces a new approach for precipitation nowcasting, aiming to maintain physically consistent trajectories over long horizons. This addresses a key limitation of current state-of-the-art neural methods, which typically rely on autoregressive approaches and suffer from error accumulation, leading to trajectory drift and physically inconsistent forecasts.

The proposed method builds on rectified flows, but replaces direct velocity prediction with prediction of the future sequence in data space. This enables an analytical expression of the induced vector field while reducing drift. The loss function further incorporates a semi-Lagrangian regularizer as a soft constraint to promote coherent transport. The method is implemented using a sophisticated, specialized backbone. Experiments are conducted on multiple datasets.

The reviewers acknowledge the importance of the problem and the novelty of the proposed method. They initially raised concerns regarding the lack of direct empirical validation of the core claim (drift reduction), the difficulty of attributing performance gains due to multiple coupled components, and limitations of the physics prior. In response, the authors provided additional diagnostics (including drift analysis), more controlled ablations, additional evaluation metrics, and clarified the role and limitations of the physical prior. These responses addressed most of the reviewers’ concerns, and all reviewers ultimately support acceptance.

The manuscript remains quite dense, and the authors are strongly encouraged to revise it to improve clarity and accessibility for a broader audience.